# The Role of Steroidomics in the Diagnosis of Alzheimer’s Disease and Type 2 Diabetes Mellitus

**DOI:** 10.3390/ijms24108575

**Published:** 2023-05-10

**Authors:** Markéta Vaňková, Marta Velíková, Daniela Vejražková, Josef Včelák, Petra Lukášová, Robert Rusina, Hana Vaňková, Eva Jarolímová, Radmila Kancheva, Josef Bulant, Lenka Horáčková, Běla Bendlová, Martin Hill

**Affiliations:** 1Institute of Endocrinology, Národní 8, 110 00 Prague, Czech Republic; mvelikova@endo.cz (M.V.); dvejrazkova@endo.cz (D.V.); jvcelak@endo.cz (J.V.); plukasova@endo.cz (P.L.); rkanceva@endo.cz (R.K.); jbulant@endo.cz (J.B.); lhorackova@endo.cz (L.H.); bbendlova@endo.cz (B.B.); 2Department of Neurology, Third Faculty of Medicine, Charles University and Thomayer University Hospital, Ruská 2411, 100 00 Prague, Czech Republic; robert.rusina@lf3.cuni.cz; 3Third Faculty of Medicine, Charles University, Ruská 2411, 100 00 Prague, Czech Republic

**Keywords:** Alzheimer’s disease, steroidome, type 2 diabetes mellitus, GC-MS, multivariate statistics, differential diagnostics

## Abstract

Epidemiological studies suggest an association between Alzheimer’s disease (AD) and type 2 diabetes mellitus (T2DM). This study aimed to investigate the pathophysiological markers of AD vs. T2DM for each sex separately and propose models that would distinguish control, AD, T2DM, and AD-T2DM comorbidity groups. AD and T2DM differed in levels of some circulating steroids (measured mostly by GC-MS) and in other observed characteristics, such as markers of obesity, glucose metabolism, and liver function tests. Regarding steroid metabolism, AD patients (both sexes) had significantly higher sex hormone binding globulin (SHBG), cortisol, and 17-hydroxy progesterone, and lower estradiol and 5α-androstane-3α,17β-diol, compared to T2DM patients. However, compared to healthy controls, changes in the steroid spectrum (especially increases in levels of steroids from the C21 group, including their 5α/β-reduced forms, androstenedione, etc.) were similar in patients with AD and patients with T2DM, though more expressed in diabetics. It can be assumed that many of these steroids are involved in counter-regulatory protective mechanisms that mitigate the development and progression of AD and T2DM. In conclusion, our results demonstrated the ability to effectively differentiate AD, T2DM, and controls in both men and women, distinguish the two pathologies from each other, and differentiate patients with AD and T2DM comorbidities.

## 1. Introduction

Aging is accompanied by an increasing number and severity of endocrine disorders. Alzheimer’s disease (AD) and type 2 diabetes mellitus (T2DM) are the most prevalent pathologies in the elderly. Both pathologies have an increasing trend, but while in AD this increase is mainly due to the gradual aging of the population, in T2DM it also depends on a sedentary lifestyle and inappropriate diet [1,2,3].

Dysregulation of steroid levels and activities of the relevant steroidogenic enzymes is associated with various neurological diseases, including AD [4]. The pathophysiology of AD is associated with amyloid-β-peptide (Aβ) deposits, hyperphosphorylated neurofibrillary tangles based on τ-protein deposits and synapse loss.

Aβ is an early-response immunopeptide, and AD patients also have elevated levels of primary stress mediators such as IL-6 and IL-1 that promote the amyloidogenic pathway [5]. Multiple steroids have been found to have the ability to protect cells from Aβ-induced death [4]. Increased AD risk has been linked to the apolipoprotein E gene (form APOEε4), and to 17-hydroxylase/C17,20-lyase (CYP17A1) polymorphisms, correlated with the age of onset of AD mainly in men [6], and Aβ can bind to sterols [7].

T2DM is a prevalent form of diabetes characterized by high blood glucose, insulin resistance (IR), relative insulin deficiency, and excessive or undue glucagon secretion. Due to the similar impairment of insulin signaling pathways and the presence of oxidative stress in AD and T2DM, sporadic AD has been suggested as a consequence of T2DM in the brain [8].

A possible link between AD and T2DM could be an impaired insulin signaling. In addition to regulating τ-protein phosphorylation in neurons, insulin signaling plays a major role in modulating synaptic function and neuronal aging.

Since AD and T2DM share some genetic and biochemical characteristics [9,10], some authors have even proposed the term “type 3 diabetes” for AD [11,12]. Both AD and T2DM are associated with hypothalamic-pituitary-adrenal axis (HPAA) defects, hypercortisolemia, and insulin resistance in the brain, leading to Aβ accumulation and τ-protein hyperphosphorylation, and consequent inflammation and neuronal loss [13].

Interestingly, regarding the comorbidity of AD and T2DM, some authors report a slower cognitive decline compared to AD patients without T2DM [14]. Moreover, others point out that although patients with T2DM may have an increased risk of developing AD, their biochemical changes and cognitive impairments are paradoxically less pronounced than in AD patients without T2DM. The term “cerebral insulin resistance” describes the inability of brain cells to respond to insulin as they normally would, resulting in impaired synaptic, metabolic, and immune function. Among the biochemical markers associated with both AD and T2DM, bioactive steroids also play an important role, both in the pathophysiology of both diseases and potentially in their diagnosis and early prediction.

In this study, we were primarily interested in the extent to which AD and T2DM are similar and different, and whether these pathologies are related only to aging or whether they are synergistic diseases associated with interacting pathophysiological cycles, and if AD and T2DM might occur in comorbidity. Since many of the observed anthropometric, biochemical, and hormonal parameters are influenced by sex, we evaluated male and female groups separately and discussed the results within and also between sex groups. We focused on steroidomic changes in groups of patients with AD and without T2DM (A+D-), T2DM without AD (A-D+), and AD with T2DM (A+D+), compared with each other and with healthy controls (A-D-).

## 2. Results

Due to the sex and age dependence for many of the variables of interest, data were assessed separately for each sex and were evaluated using age-adjusted ANCOVA, followed by Bonferroni multiple comparisons as well as the orthogonal projections to latent structure (OPLS) models. Age-adjusted ANCOVA assessed the effects of AD, T2DM, and AD and T2DM comorbidity. Bonferroni multiple comparisons evaluated the differences between all group combinations (*p* < 0.05). The OPLS models were focused on the distinctions between pairs of groups.

### 2.1. Anthropometric and Biochemical Parameters

AD patients (A+D- and A+D+) were older than the other compared groups of T2DM patients (A-D+) and controls (A-D-). There was no significant difference in age between T2DM patients (A-D+) and controls (A-D-). The obesity-related anthropometric parameters, such as BMI, WHR, BAI, abdominal, waist, and hip circumference, and systolic blood pressure, were significantly higher in T2DM (A-D+) compared to controls (A-D-). Patients with AD (A+D-) compared to T2DM (A-D+) had significantly lower values of anthropometric parameters, similar to the control group (A-D-). In anthropometric parameters, the group with a combination of both T2DM and AD diagnoses (A+D+) was more similar to the AD (A+D-) and control groups (A-D-). The results suggest antagonism between T2DM and AD in anthropometric parameters (see T2DM factor vs. AD factor, Table 1 and Table 2).

Concerning glucose metabolism, glucose levels showed a negative association with AD (AD factor) but a positive association with T2DM (T2DM factor) in both sexes (Table 1 and Table 2). Women and men with T2DM (A-D+) had significantly higher glycemia compared to controls (A-D-) and compared to AD patients (A+D-) as well. C-peptide and proinsulin levels were also positively associated with T2DM. The effect of AD on the insulin resistance index (HOMA R) was not demonstrated, and HOMA R in both sexes positively correlated only with T2DM. On the other hand, the value of the insulin secretion index (HOMA F) was higher in both sexes in AD patients (A+D-) than in T2DM (A-D+).

The relationship between lipid parameters and AD and T2DM was complex and varied by sex. Lipid parameters (Table 1 and Table 2) were significantly associated with AD and T2DM only in women. Free fatty acids, total cholesterol, and LDL cholesterol were positively influenced by AD (AD factor), and the HDL cholesterol level was negatively influenced by T2DM (T2DM factor). However, a significant proportion of patients in all groups were taking hypolipidemic therapy.

As for liver function, GGT was not associated with either AD or T2DM in either sex (Table 1 and Table 2). The hepatic transaminases ALT and AST were negatively correlated with AD (A+D-) in both sexes, while the AST/ALT ratio showed a positive correlation with AD. In contrast, elevated ALT levels and a decreased AST/ALT ratio were observed in T2DM patients (A-D+). Patients with both diagnoses (A+D+) had liver enzyme results and an AST/ALT ratio more similar to those with AD (A+D-) than to those with T2DM (A-D+).

In men, thyroid hormones had no association with AD or T2DM. In women, free thyroxin and free triiodothyronine levels were negatively influenced by AD (AD factor).

Regarding markers of renal function, elevated uric acid levels were associated with the presence of T2DM. In AD patients (A+D-), uric acid levels were reduced compared to T2DM (A-D+). Uric acid levels were significantly lower in the group with a combined diagnosis (A+D+) compared to the group with T2DM only (A-D+). The creatinine levels were within the physiological range in all groups, with the lowest levels in the A+D+ group.

The only inflammatory parameter monitored was C-reactive protein (CRP). No significant association of CRP with AD or T2DM was found in women, and a borderline association of CRP with AD (negative correlation) was suggested in men.

### 2.2. Sex Hormone-Binding Globulin (SHBG) and Steroids

Thanks to the detailed analysis of the steroidome in all subjects, statistical analysis (ANCOVA) revealed many significant differences in steroid levels between the study groups, which were particularly pronounced in women (Table 3 and Table 4). The differences found are complemented by the OPLS models, see chapter 2.3 (OPLS Models and Steroids).

#### 2.2.1. SHBG

Positive relationships between SHBG and AD were observed in non-T2DM subjects (A+D-) of both sexes. SHBG was negatively correlated with T2DM in women without AD (A-D+), but not in men. Patients with AD (A+D-) had significantly higher SHBG levels than patients with T2DM (A-D+) in both sexes.

#### 2.2.2. Δ^5^ Steroids

In women, AD patients (A+D-) and T2DM patients (A-D+) had elevated levels of most Δ^5^ C21 steroids (pregnenolone sulfate, 17-hydroxypregnenolone, 16α-hydroxypregnenolone) compared to controls (A-D-). Moreover, women with diabetes had higher levels of pregnenolone and 20α-dihydropregnenolone sulfate compared to control women. Regarding Δ^5^ C19 steroids, AD negatively influenced the levels of androstenediol sulfate and 5-androstane-3β,7β,17β-triol. The combined group (A+D+) had significantly lower levels of these steroids than diabetic women (A-D+). Surprisingly, multivariate analysis revealed no differences in dehydroepiandrosterone (DHEA) and DHEA sulfate among groups of both sexes.

However, OPLS models comparing women with AD (A+D-) vs. controls (A-D-) revealed a positive correlation of AD diagnosis with DHEA and its derivatives, 7α-hydroxy-DHEA and 16α-hydroxy-DHEA sulfate. This was not seen in men with AD. T2DM did not correlate with Δ^5^ C19 steroids, with the only exception being 16α-hydroxy-DHEA sulfate, which slightly positively correlated with T2DM in women (Table 3 and Table 4).

#### 2.2.3. Progestogens

Women with AD (A+D-) had significantly higher levels of 17-hydroxyprogesterone compared to controls (A-D-) and even to T2DM (A-D+). Women with T2DM (A-D+) had elevated progesterone levels vs. controls (A-D-), vs. patients with AD (A+D-), and even vs. combined diagnosis (A+D+). In men, AD (A+D-) was also associated with higher 17-hydroxyprogesterone compared to T2DM (A-D+). Similar to women, the highest levels of progesterone were seen in male patients with T2DM (A-D+), and the highest levels of 17-hydroxyprogesterone in patients with AD (A+D-) (Table 3 and Table 4).

#### 2.2.4. Cortisol

Cortisol levels were positively correlated with AD, but they were independent of T2DM in both sexes. AD patients (A+D-) had the highest levels of cortisol within the groups and significantly differed from controls (A-D-), and in women, also differed from diabetic patients (A-D+) (Table 3 and Table 4).

#### 2.2.5. Androstenedione and Active Androgens

Androstenedione was elevated both in patients with AD (A+D-) and T2DM (A-D+) compared to controls (A-D-), but only in women. The groups did not differ in active androgens such as testosterone and 5α-dihydrotestosterone. The OPLS model revealed a positive correlation of 5α-dihydrotestosterone with T2DM in women but was negative in men (Table 3 and Table 4).

#### 2.2.6. Estrogens

In both sexes, estradiol positively correlated with T2DM (factor T2DM). The patients with T2DM (A-D+) had higher levels than controls (A-D-), AD patients (A+D-), and patients with a combined diagnosis (A+D+). In women, T2DM patients (A-D+) had significantly higher estrone levels than controls. Estrogens were strong predictors in the OPLS models discriminating T2DM patients (Table 3 and Table 4).

#### 2.2.7. 5α/β-Reduced C21 and C19 Steroids

The results indicate that patients with AD (A+D-) and patients with T2DM (A-D+) had elevated levels of many 5α/β-reduced C21 and C19 steroids compared to controls (A-D-) (Table 3 and Table 4). These changes were pronounced in women with AD as well as in women and men with T2DM. When we compared patients with T2DM (A-D+) to patients with AD (A+D-), diabetic men had higher allopregnanolone, isopregnanolone, 5α-androstane-3α,17β-diol and its conjugate, and conjugated 5α-androstane-3β,17β-diol. Women with T2DM (A-D+) had higher 5α-androstane-3α,17β-diol compared to AD patients (A+D-). The group of combined diagnoses (A+D+) did not differ from controls (A-D-) in any of the screened steroid parameters. However, this group (A+D+) had significantly lower levels of isopregnanolone, 5α-androstane-3α,17β-diol, and in women, also their conjugated forms, compared to the T2DM group (A-D+).

### 2.3. OPLS Models and Steroids

#### 2.3.1. Distinguishing Volunteers with AD without T2DM (A+D-) from Controls (A-D-)

The OPLS model to distinguish women with AD without T2DM (A+D-) from female controls (A-D-) showed predominantly positive associations with numerous pregnane and androstane steroids and SHBG (Table 5). The OPLS model explained 61.2% (53.4% after cross-validation) of the total variability in the explained variable, and discrimination was highly effective (sensitivity = 0.923 (0.821, 1), specificity = 0.902 (0.812, 0.993), shown as means with 95% confidence intervals). Similarly, the OPLS model to distinguish men with AD without T2DM (A+D-) from male controls (A-D-) showed mostly positive associations of AD with several pregnane and androstane steroids and SHBG (Table 6). The number of relevant steroids was lower, but the effectiveness of discrimination was similar to that of women. The OPLS model explained 59.4% (56.5% after cross-validation) of the total variability in the explained variable (sensitivity = 0.923 (0.821, 1), specificity = 0.833 (0.661, 1)).

#### 2.3.2. Distinguishing Volunteers without AD with T2DM (A-D+) from Controls (A-D-)

The OPLS model to distinguish women without AD with T2DM (A-D+) from female controls (A-D-) showed exclusively positive correlations of T2DM with various steroids, but inverse correlations with SHBG. The OPLS model explained 81.4% (73.3% after cross-validation) of the total variability in the explained variable (sensitivity = 1 (1, 1), specificity = 0.941 (0.829, 1)) (Table 7). Similarly, the OPLS model to distinguish men without AD with T2DM (A-D+) from male controls (A-D-) showed mostly positive associations of T2DM with various steroids, except for 17-hydroxypregnenolone, which showed an inverse correlation. The number of relevant steroids in men was lower than that in women, but not the effectiveness of discrimination. The OPLS model explained 75.9% (69.1% after cross-validation) of the total variability in the explained variable (sensitivity = 1 (1, 1), specificity = 0.941 (0.829, 1)) (Table 8).

#### 2.3.3. Discrimination of Volunteers with AD without T2DM (A+D-) from Volunteers without AD with T2DM (A-D+)

The OPLS model to distinguish women with AD without T2DM (A+D-) from women without AD with T2DM (A-D+) showed a positive correlation with SHBG and positive and negative correlations with a number of steroids. The discriminatory power of the OPLS model was high as it explained 80.7% (72.9% after cross-validation) of the total variability in the explained variable (sensitivity = 0.963 (0.892, 1), specificity = 0.978 (0.936, 1)) (Table 9). The OPLS model to distinguish men with AD without T2DM (A+D-) from men without AD with T2DM (A-D+) also showed positive correlations with SHBG and positive and negative correlations with different steroids, and the discriminatory power of the OPLS model, which explained 84.8% (77.4% after cross-validation) of the total variation in the explained variable, was absolute (Table 10).

#### 2.3.4. Distinguishing Volunteers with AD with T2DM (A+D+) from Volunteers with AD without T2DM (A+D-)

The OPLS models to distinguish women with AD with T2DM (A+D+) from women with AD without T2DM (A+D-) showed exclusively negative correlations of combined diagnosis (A+D+) with multiple steroids. However, the effectiveness of the discrimination was very low because the OPLS model explained only 27.6% (17.6% after cross-validation) of the total variability in the explained variable (sensitivity = 0.167 (0, 0.465), specificity = 1 (1, 1)) (Table 11). The OPLS model to distinguish men with AD with T2DM (A+D+) from men with AD without T2DM (A+D-) also showed exclusively inverse correlations of A+D+ with several steroids, and the effectiveness of discrimination was also low because the OPLS model explained only 46.5% (37.5% after cross-validation) of the total variability in the explained variable (sensitivity = 0.429 (0.062, 0.795), specificity = 1 (1, 1)) (Table 12).

#### 2.3.5. Distinguishing Volunteers with AD with T2DM (A+D+) from Volunteers without AD with T2DM (A-D+)

In contrast to the situation in non-diabetic women, the OPLS model to distinguish women with AD with T2DM (A+D+) from women without AD with T2DM (A-D+) showed mostly negative associations of A+D+ with many steroids, with the exception of 17-hydroxyprogesterone, which was positively correlated with a combined diagnosis (A+D+). The OPLS model explained 74.6% (60.5% after cross-validation) of the total variability in the explained variable, and discrimination was absolute (Table 13). The OPLS model to distinguish men with AD with T2DM (A+D+) from men without AD with T2DM (A-D+) showed a mostly inverse association of AD with various steroids, with the exception of pregnanolone, which was positively correlated with a combined diagnosis (A+D+). However, the number of relevant steroids was lower than that in women, as was the effectiveness of discrimination. The OPLS model explained 60.9% (48.3% after cross-validation) of the total variability in the explained variable (sensitivity = 0.833 (0.535, 1), specificity = 0.960 (0.883, 1)) (Table 14).

## 3. Discussion

Our study aimed to investigate to what extent AD and T2DM are similar, how they differ, and how they influence each other. We focused on the differences in anthropometric and biochemical parameters, especially the steroid levels and their metabolites. Specifically, we evaluated biochemical and hormonal changes in groups of patients with AD and without T2DM (A+D-), T2DM without AD (A-D+), and AD with T2DM (A+D+), compared with each other and with healthy controls (A-D-).

Data were evaluated using two independent statistical approaches (ANCOVA, OPLS), and the most significant results regarding the pathogenesis of AD and T2DM are discussed below. Using OPLS models, we also attempted to distinguish the study groups of patients from each other and the control group.

### 3.1. Anthropometric and Biochemical Parameters

Patients with AD (A+D-) were older than the groups of T2DM patients (A-D+) and controls (A-D-), and therefore all the observed parameters were adjusted for age. Data from both ANCOVA and OPLS models showed that obesity-related anthropometric parameters, such as BMI, WHR, BAI, abdominal, waist, and hip circumference, and systolic blood pressure, were mostly significantly positively correlated with T2DM (A-D+), whereas lower values of anthropometric parameters were significantly associated with AD (A+D-), especially in women. Patients with both diagnoses (A+D+) were more similar in anthropometric parameters to patients with AD (A+D-) than to those with T2DM (A-D+). These findings are consistent with the literature data. Obesity is a hallmark of T2DM, and AD is associated with weight loss and under-nutrition [15,16]. Disturbed eating behaviors and reductions in body weight may be seen nearly two decades before the diagnosis of AD [17]. A large retrospective study in 2015 involving nearly 2 million participants [18] showed that being underweight in middle and older age is associated with an increased risk of dementia over two decades.

Impaired glucose metabolism in the brain of AD patients is evident early in the disease and is associated with insulin resistance with concomitant hyperinsulinemia, features common to T2DM and AD. Circulating insulin crosses the blood–brain barrier and can affect glucose metabolism in the brain, act on the removal of Aβ, increase the phosphorylation of τ-proteins, and increase the concentration of pro-inflammatory substances in the brain (for a review, see [19]).

Our data showed that AD patients of both sexes, regardless of T2DM, had lower glycemia compared to T2DM patients and controls, suggesting a glucose deficit in AD patients already at a peripheral level. However, AD patients had lower levels of fasting beta-cell secretion markers, compared to T2DM patients, especially in women, but comparable to controls. When directly comparing the two pathologies (without patients with comorbidities), AD patients showed a significantly higher insulin secretion index (HOMA F) compared to controls and even to T2DM patients. This fact may be important for understanding the pathophysiology of AD since a higher insulin to glucose ratio in AD patients with unchanged or even lower insulin resistance leads to lower glucose levels. Reduced peripheral glucose levels may ultimately contribute to cerebral malnutrition in AD patients, a hypothesis proposed by some authors [20]. Thus, our present data are consistent with the idea of hyperinsulinemia (even relative) as the cause of glucose deficiency in the brain of AD patients.

Abnormal lipid metabolism in the brain and periphery has been implicated in the pathophysiology of AD [21,22,23,24]. Cholesterol, which serves as a precursor of endogenous steroids, is retained in the brain of AD patients, and the metabolic disorder is associated with changes in β- and γ-secretase activity (see [19] for a review). Increased levels of free fatty acids (FFA) in plasma induce insulin resistance and thus play a key role in the development of T2DM [25]. In contrast to our current data showing a positive correlation between total cholesterol and triglycerides with AD in women, Kuusisto et al. [26] reported decreased total cholesterol but unchanged triglycerides in AD patients. However, Akyol et al. [24] reported that diacylglycerols along with triacylglycerols were the most significantly elevated lipids when comparing AD and control brains, and these more recent data are consistent with our present results in peripheral blood. In the study by Kuusisto et al. [26], HDL cholesterol in subjects who developed AD did not differ from controls, as in our current data. In our study, the levels of lipid parameters, probably due to considerable hypolipidemic treatment in the volunteers, were not significantly different in AD patients (A+D-) compared to controls (A-D-) or compared to T2DM (A-D+).

One of the key functions of the liver is metabolic detoxification, which also includes peripheral metabolic clearance of Aβ. Peripheral clearance of Aβ facilitates the outflow of Aβ from the brain, and therefore, inadequate peripheral clearance of Aβ due to impaired liver function contributes to the progression of AD [27]. Other authors [28,29] have reported that an increased AST/ALT ratio and lower ALT levels are associated with the diagnosis of AD, and these changes are related to increased Aβ deposition, higher levels of phosphorylated and total τ-protein in cerebrospinal fluid, as well as decreased cerebral glucose metabolism and greater brain atrophy. Our present data showed that, in addition to unchanged GGT levels, hepatic transaminases (ALT and AST) were decreased in AD patients, while the AST/ALT ratio was increased, supporting the idea of a link between impaired liver function on the one hand and lower Aβ clearance, increased Aβ deposition, and decreased brain glucose metabolism on the other [27,28,29].

Dysregulation of thyroid hormones can significantly affect metabolism. Triiodothyronine regulates cholesterol metabolism by influencing gene expression and interacting with other nuclear receptors, and it modulates hepatic insulin sensitivity by inhibiting hepatic gluconeogenesis. Both fT3 and TSH levels decrease with age, while the incidence of autoimmune thyroid disease increases with age, and these changes may be related to several pathologies, including T2DM and AD [30]. The transport protein transthyretin, which selectively binds thyroid hormones, binds Aβ and reduces its concentration in the cerebrospinal fluid of AD patients (for a review, see [19]). A negative association between serum fT4 levels and global Aβ deposition in the brain was found even after adjustment for the effects of age, sex, and APOEε4 genotype, even in the clinically euthyroid state [31]. Quinlan et al. [32] reported that fT4 levels were higher in AD patients, and at the same time, fT3 levels were positively correlated with left amygdala volume and, in controls, T3 levels were positively correlated with hippocampal volume, suggesting a protective effect of T3 and fT3 in AD patients. In our present data, thyroid hormones were significantly correlated only in women. Free thyroxine was higher in women with T2DM (A-D+), and free triiodothyronine was lower in women with AD (A+D-). The protective effect of thyroid hormones was attenuated in women with comorbidity of AD and T2DM (A+D+).

Uric acid is the end-product of purine oxidation in the circulation. Low uric acid levels may contribute to oxidative stress, which accelerates AD progression [20]. Some authors [20,33] have suggested that low serum uric acid levels may be related to AD-related brain hypometabolism, and that both low uric acid levels and AD progression may be caused by malnutrition. Most studies report a negative relationship between uric acid levels and AD [34,35,36,37]. In our study, uric acid levels were elevated only in patients with T2DM (A-D+). Patients with AD (A+D-) had uric acid levels that were lower compared with T2DM but comparable to controls, and uric acid levels in AD patients and controls were within the physiological range.

C-reactive protein (CRP) is an acute-phase effector that has been associated with AD in histopathological and longitudinal studies. Some studies have reported that low basal plasma CRP levels were associated with an increased risk of AD [38,39]. However, our present results did not demonstrate a significant effect of AD or T2DM on CRP levels.

### 3.2. Sex Hormone-Binding Globulin (SHBG) and Steroids

#### 3.2.1. SHBG

SHBG is a glycoprotein that binds to androgens and estrogens, inhibits their activity, and serves as a carrier and reservoir for their future use. In addition to insulin, SHBG synthesis is inhibited by testosterone and prolactin and stimulated by estradiol [40]. Most SHBG is formed in the liver and released into the bloodstream with a half-life of one week. In addition, SHBG is also produced in the brain and may influence the progression of AD. SHBG levels are approximately twice as high in women as in men [41,42]. Data on the association between SHBG, its internalization in the brain, and Aβ clearance have been published [41,43]. Studies investigating the relationship between SHBG and AD have found mostly positive correlations between AD, disease progression, or the risk of developing AD [44,45,46,47,48]. On the other hand, other studies have found no changes in SHBG levels in AD patients [41], as this relationship is complicated by the general increase in SHBG levels with age. Genetic studies also provide evidence that SHBG is involved in the etiology of T2DM [49]. Studies examining the relationship between SHBG and T2DM have generally reported an inverse relationship between the prevalence and/or risk of T2DM [50,51,52,53,54].

In our data, the AD factor had an effect on increased SHBG levels in men and the T2DM factor had an effect on decreased SHBG levels in women, which is consistent with the above studies. When comparing groups, AD patients (A+D-) of both sexes had higher SHBG levels than T2DM patients (A-D+) and controls (A-D-).

#### 3.2.2. Δ^5^ Steroids

In addition to the high production of active androgens in the male testes, the adrenal cortex is an important source of most steroids in older people. Other sources of steroids include other peripheral tissues, especially adipose tissue, and, to a limited extent, cells of the nervous system. Additionally, steroids in the major Δ^5^ pathway are predominantly produced in the adrenal cortex, which consists of three zones with specific steroid production, each zone being controlled by a different enzyme system (see the reviews in [55,56]).

Regarding our results from comparing groups, we found significant differences in Δ^5^ C21 and C19 steroid levels, exclusively in women. Women with AD (A+D-), as well as women with T2DM (A-D+), had elevated levels of all the Δ^5^ C21 steroids we studied compared with controls (A-D-), but there were no differences between AD and T2DM patients (with one exception: higher levels of 17-hydroxypregnenolone in AD).

Pregnenolone, which is an early precursor of bioactive steroids, usually counteracts the development of AD and other neuropathologies, and its elevated levels can be interpreted as a counter-regulatory protective mechanism [7]. However, the findings of studies that have investigated the relationship between pregnenolone and T2DM are inconsistent [57,58,59,60]. In our study, we found elevated pregnenolone levels in both women with AD (A+D-) and women with T2DM (A-D+).

Pregnenolone sulfate is a weak positive modulator of the N-methyl-D-aspartate receptor (NMDAR). Similar to the α-amino-3-hydroxy-5-methyl-4-isoxazolepropionic acid receptor (AMPAR), NMDARs are essential for synapse integrity, modulation of synaptic plasticity, and subsequently, for the functioning of spatial memory and learning, and NMDAR functionality and the expression of its subunits in the nervous system decline with increasing age (for reviews, see [4,56]). Mayo et al. [61] reported that pregnenolone sulfate improves cognitive function in the brains of laboratory animals. Pregnenolone sulfate is also a positive modulator of Kir2.3 ion channels (potassium inwardly rectifying channel, subfamily J, member 4), which play an important role in cognitive quality, memory, emotionality, and may also be involved in the pathophysiology of some neuropsychiatric diseases (see the review in [56]). In postmortem samples of AD patients, pregnenolone sulfate levels in the temporal cortex correlated with the neuropathological stage of the disease, and there was a trend towards higher pregnenolone sulfate levels in AD patients compared with cognitively healthy controls [62]. In addition, toxic doses of Aβ significantly increased pregnenolone sulfate levels in cultured SH-SY5Y (in vitro models of neuronal function and differentiation) cells in a time-dependent manner [4]. Our present results, i.e., higher pregnenolone sulfate levels in women with AD, were consistent with our previous study [63], and were also consistent with the concept of pregnenolone sulfate involvement in a counter-regulatory mechanism to overcome the deleterious effects of AD (see the review in [4]). In contrast, our current data on elevated pregnenolone sulfate levels in women with T2DM (A-D+) were not consistent with the results of the study by Tagawa et al. [57], who reported lower pregnenolone sulfate levels in diabetic compared to non-diabetic patients.

In women with AD (A+D-) and T2DM (A-D+), the level of 17-hydroxypregnenolone was also positively correlated. The 17-hydroxypregnenolone is a key precursor in the metabolic pathway leading to cortisol synthesis, and cortisol is known to be a diabetogenic steroid. However, cortisol levels in our study were positively correlated only with AD, not with T2DM.

Several studies, including one of our previous papers [64], suggest that the imbalance between DHEA and DHEAS in the brain induced by Aβ [4] plays an important role in the pathogenesis of AD. While DHEA levels in the central nervous system have been described to be elevated, DHEAS levels are conversely reduced [64] (also see the review in [65]). In general, DHEA has protective effects against cellular and in vivo toxicity induced by Aβ [7]. Weill-Engerer et al. [66] observed a general trend of decreasing levels of all steroids, including DHEA and DHEAS, in brain regions of AD patients compared to controls. Similarly, Schumacher et al. reported [67] that AD patients tend to have reduced levels of neurosteroids in various brain regions, with neurosteroid levels inversely correlated with phosphorylated τ-protein and Aβ. However, other studies have reported high DHEA concentrations in the prefrontal and temporal cortex in postmortem samples of AD patients (see the review in [4]). In addition, Naylor et al. [62] reported elevated DHEA levels in the cerebrospinal fluid of AD patients compared to cognitively intact controls.

Regarding DHEA and DHEAS levels in the peripheral circulation of AD patients, a meta-analysis [68] found no statistically significant association between DHEA levels in AD patients but lower DHEAS levels compared to controls. Cho et al. [69] found significantly reduced DHEAS concentrations in AD patients compared to control women. In another study, AD patients had lower DHEAS levels than older patients without dementia [70]. Ray et al. [71] also found lower DHEAS levels in AD patients compared to controls, but this study did not account for gender differences. Hayashi et al. [72] found significantly lower DHEA levels in men with AD, but no difference in women. Ponholzer et al. [73] found that AD progression in a group of 75-year-old men was associated with a decrease in serum DHEAS levels. Regarding DHEA levels, Bernardi et al. [74] showed that AD patients have lower DHEA levels compared to controls. In summary, peripheral DHEAS levels were negatively correlated with AD in most studies, but data on DHEA levels were inconsistent.

A comparison of the groups in our current study showed, in contrast to the cited studies, increased DHEA levels in women with AD (A+D-) compared to controls and women with T2DM (A-D+), and increased levels of other Δ^5^ C19 steroids, such as 7α-hydroxy-DHEA and 16α-hydroxy-DHEA sulfate, compared to controls. Surprisingly, we found no differences in DHEAS levels in our cohorts.

The DHEA metabolite, androstenediol, is a neuroprotective steroid that reduces axon damage caused by demyelination, presumably by reducing the local inflammatory response in the white matter [75]. Androstenediol is one of the precursors of the estradiol pathway and is itself active at both types of estrogen receptors. Moreover, androstenediol and its metabolite, 5-androstene-3β,7β,17β-triol, are immuno-protective steroids [76]. The 5-androstene-3β,7β,17β-triol, which can be formed either by interconversion from 5-androstene-3β,7α,17β-triol or directly from androstenediol by the catalytic action of cytochromes CYP3A4 and CYP3A7, is a highly potent immuno-protective steroid, despite its low concentration due to high clearance (summarized in [77]). In our cohort, we observed no significant differences in these steroids between women with AD and T2DM compared to controls. Only women in the combined group (A+D+) had reduced levels of both androstenediol sulfate and both triols (5-androstene-3β,7α,17β-triol, -androstene-3α,7α,17β-triol), and these negative correlations were rather unfavorable for the group with comorbidity of AD and T2DM (A+D+).

#### 3.2.3. Progestogens

Progesterone circulating at sub-nanomolar levels, common in men and postmenopausal women, is predominantly adrenal in origin (see the review in [78]). Due to its lipophilicity, it readily crosses the blood–brain barrier into the central nervous system and is active in both the central and peripheral nervous systems (see the review in [78]). Progesterone also promotes glucose metabolism in neurons by increasing the expression of glucose transporters [79]. Branisteanu and Mathieu [80] reported that progesterone decreases glucose uptake and increases glucose release from the liver. At low insulin levels, progesterone inhibits glucose uptake, stimulates hepatic glucose production, and may contribute to maintaining circulating glucose levels.

A comparison of the groups shows that in women, T2DM patients (A-D+) have the highest progesterone levels, while controls (A-D-) and women with AD (A+D-) have similar levels. This would suggest an increased protective effect of progesterone in T2DM (A-D+), but not in the combined AD and T2DM group (A+D+), in which the protective effect of progesterone appeared to fail. Our results are in agreement with data from Jiang et al. [59], who reported higher progesterone levels in patients with T2DM. On the other hand, Liu et al. [60] found no association of T2DM with progesterone levels.

In our study, women with AD (A+D-) had the highest 17-hydroxyprogesterone levels compared with controls (A-D-) and women with T2DM (A-D+), which was consistent with our previous study [63]. In men, 17-hydroxyprogesterone levels were higher in AD patients (A+D-) compared to T2DM patients (A-D+). In contrast to our results, Hayashi et al. [72] found no significant differences for 17-hydroxyprogesterone in both sexes. Lu et al. [81] described elevated 17-hydroxyprogesterone levels in patients with T2DM. On the other hand, Liu et al. [60] found no significant association of T2DM with 17-hydroxyprogesterone (regardless of gender differences).

#### 3.2.4. Cortisol

Cortisol is a key immunosuppressive glucocorticoid that affects glucose homeostasis, and dysregulation of the hypothalamic-pituitary-adrenal axis (HPAA) is an important link between stress and T2DM. Elevated cortisol levels and overall HPAA dysregulation are also risk factors for the progression and development of AD (see the reviews in [19,82]). Cortisol generally induces hyperglycemia and insulin resistance, increases Aβ production and τ-protein hyperphosphorylation, decreases brain neuroplasticity, and induces hippocampal atrophy and memory loss (see the review in [5]). Administration of glucocorticoids at levels corresponding to the stress response increased Aβ production and τ-protein accumulation and accelerated neurofibrillary tangle development [83]. While a group of healthy subjects responded by increasing plasma glucose levels after cortisol stimulation, a group of age-matched AD patients did not (see the review in [19]), suggesting a reduced sensitivity to glucocorticoids in AD patients in terms of glucose mobilization. This phenomenon may be related to malnutrition in these patients already at the peripheral level. Some authors have suggested that high glucocorticoid levels in AD are not a consequence of the disease but rather play a central role in the development and progression of AD [83]. However, there are also studies that have found no difference in cortisol levels between AD patients and matched controls in either sex [72].

When comparing groups, both men and women with AD (A+D-) had much higher cortisol levels than both controls and T2DM patients (A-D+), but this was not true for patients with AD and T2DM at the same time (A+D+). We speculate that the high cortisol levels in our AD patients may be due to a stress response associated with the manifestations of the early phase of their disease (see the Section 4). Although a number of studies have reported positive correlations between cortisol on the one hand, and T2DM and insulin resistance on the other [60,84,85,86], our data did not support these findings in either sex.

#### 3.2.5. Androstenedione and Active Androgens

Literature data suggest that at least testosterone in men functions as a protective agent promoting Aβ removal, suppressing inflammation, and regulating insulin signaling and synaptic plasticity in the brains of AD patients (see the review in [4]). While Ponholzer et al. [73] reported that serum testosterone levels in men were not related to the prevalence or incidence of AD, Hayashi et al. [72] found significantly lower testosterone levels in men with AD, but no significant difference in women. However, in our present data, testosterone levels in both sexes were not related to AD, which was consistent with our previous study [63].

A comparison of the groups showed that women with T2DM (A-D+) had higher 5α-dihydrotestosterone levels than women with AD (A+D-), while the opposite was true for men, with AD patients having higher levels than diabetic patients. Group comparisons further showed that androstenedione, the immediate precursor of testosterone and estrone [87], was higher in women with AD (A+D-) and in both men and women with T2DM (A-D+) compared to controls. No significant association with AD was found in men. In women with AD without diabetes (A+D-), our present results were consistent with our previous study [63]. In contrast, Hayashi et al. [72] found lower androstenedione levels in AD patients (A+D-) compared to controls in both sexes. Higher serum androstenedione levels in women with T2DM were described by Tok et al. [88]. In addition, Diboun et al. [58] reported higher androstenedione levels in women with insulin resistance compared to insulin-sensitive women.

#### 3.2.6. Estrogens

Estrogen precursors in older men and women are predominantly of adrenal origin, with peripheral tissues, particularly adipose tissue, being the main source of active estrogens [89] (also see the reviews in [55,56]).

With respect to AD, estradiol reduces τ-protein hyperphosphorylation and regulates excitatory synaptic transmission in hippocampal neurons via estrogen receptor activation. In addition, estradiol increases the expression of the brain transport protein transthyretin, which selectively binds thyroid hormones and additionally binds Aβ and decreases its concentration in the cerebrospinal fluid of AD patients (see the reviews in [4,19]). Genes involved in estrogen biosynthesis and estrogen receptor activity may contribute to the AD risk by influencing the age of onset of AD and by affecting the neuroprotective activity of estrogens (see the review in [90]). The loss of estradiol at menopause could be responsible for the increased risk of AD in women [91] (also see the review in [4]). Estradiol is therefore mostly associated with beneficial effects in AD.

Literature data on the relationship of estradiol and estrone levels to T2DM are discrepant [58,88,92,93] (also see the review in [43]). In summary, estradiol in T2DM promotes insulin secretion, inhibits glucagon release and hepatic gluconeogenesis, promotes pancreatic β-cell survival, stimulates energy expenditure and food intake, and promotes glucose uptake in muscle and adipocytes. However, available data on the relationship between estrogen levels and T2DM are inconsistent.

A comparison of the groups showed that in both males and females, estradiol levels were highest in T2DM patients (A-D+), while AD patients (A+D-) of both sexes did not significantly differ from controls in estrogen levels. Our study did not confirm an association of estrogen levels with AD in either men or women. On the contrary, similar to some of the studies mentioned above, we confirmed positive correlations of estrogens with T2DM. Higher estradiol levels are probably related to a higher proportion of adipose tissue in patients with T2DM.

#### 3.2.7. 5α/β-Reduced C21 and C19 Steroids

The various 5α/β-reduced metabolites of progesterone and androstenedione are neuroprotective substances that have anti-inflammatory effects, stimulate myelination and remyelination of Schwann cells in the peripheral nervous system, protect mitochondria, regulate neurogenesis, affect mood, memory, and cognition (see the reviews in [4,78]), and protect nervous system cells from hyperexcitation, a factor contributing to the onset and development of AD [94]. The 5α/β-reduced C21 and C19 steroids are active at a number of receptors, including GABAA (γ-aminobutyric acid type A) and glutamate receptors (see the review in [78]). One of these steroids, allopregnanolone, counteracted Aβ-induced neurotoxicity independently of the GABAA receptor through suppression of Aβ-induced phosphorylation of an extracellular signal-regulated kinase (see the review in [4]). In addition, allopregnanolone suppressed τ-protein expression in rat brains [95]. The 5α/β-reduced steroids may be involved in the pathophysiology of both T2DM and AD (see the review in [56]), although results regarding changes in the levels of some of these steroids in AD-affected brains are conflicting. Marx et al. [96] reported that allopregnanolone levels were significantly lower postmortem in the prefrontal cortex of AD patients than in controls.

Pancreatic β-cells produce significant amounts of GABA (γ-aminobutyric acid), which activates the GABAA receptor and subsequently inhibits glucagon secretion in α-cells. This opens the possibility of implicating GABAergic reduced progesterone metabolites such as allopregnanolone and pregnanolone, and possibly analogous reduced C19 steroids such as androsterone, epiandrosterone, and 3α-hydroxy-5α/β-androstanediols, in the pathophysiology of T2DM. Thus, it is possible that the aforementioned steroid metabolites may also influence glucose homeostasis and thus the onset and progression of T2DM (see the review in [78]). The study by Afrazi et al. [97] described a protective effect of allopregnanolone against proapoptotic pathologies, including T2DM. The authors suggested that this neuroactive steroid may ameliorate some of the adverse effects of diabetes, such as diabetic neuropathy.

A comparison of the groups showed that both AD patients (A+D-) and T2DM patients (A-D+) had elevated levels of many 5α/β-reduced C21 and C19 steroids compared to controls (A-D-). When AD and T2DM patients were compared with each other, the levels of these steroids were almost exclusively higher in T2DM patients (A-D+).

The combined diagnosis group (A+D+) did not differ from controls (A-D-) in any of the steroid parameters studied, but they had significantly lower levels of some C21 and C19 5α/β-reduced steroids than both T2DM (A-D+) and AD (A+D-) patients. Our present results regarding both C21 and, to a lesser extent, 5α/β-reduced C19 steroids suggest a compensatory mechanism by which increasing levels of these predominantly neuroprotective agents may attenuate the development and progression of AD.

Our data also suggest that 5α/β-reduced C21 steroids are involved in counter-regulatory mechanisms that protect against some of the adverse consequences of T2DM. However, there is one unfavorable aspect regarding the possible involvement of the aforementioned steroids in the pathophysiology of T2DM. Persistently elevated levels of GABA and GABAergic steroids, which stimulate food intake and weight gain in T2DM patients, may contribute to the development and progression of T2DM.

## 4. Materials and Methods

### 4.1. Subjects

A total of 88 AD patients, 48 women, and 40 men, who fulfilled The National Institute of Neurological and Communicative Disorders and Stroke and the Alzheimer’s Disease and Related Disorders Association (NINCDS-ADRDA) criteria for probable AD, as well as a total of 86 T2DM patients, 54 women and 32 men, and a total 59 healthy controls over 65 years, 41 women and 18 men, participated in the study. Among the AD patients, 14 patients also had T2DM, and these patients with a combination of AD and T2DM diagnoses (7 women and 7 men) were counted as a separate group. In our AD cohort, 67 patients had AD, 7 patients had possible dementia with Lewy bodies in comorbidity, and 11 patients had AD and significant subcortical ischemic white matter lesions, consistent with the diagnosis of mixed dementia.

Participants were examined after signing an informed consent form approved by the Ethics Committee of the Institute of Endocrinology. For the evaluation of basic biochemical parameters and steroid metabolome, the peripheral blood was withdrawn after fasting in the morning. Blood samples were centrifuged and stored at −20 °C until analysis.

This study had several limitations. First, the group of AD patients was significantly older than the group of T2DM patients and controls. Most parameters studied were age-dependent, so we used age correction in the analyses. Second, the group of patients with a combined diagnosis of AD and T2DM, although numerically small, was very interesting; therefore, we decided to keep it in the analysis.

### 4.2. Screening Tools for Estimation of Cognitive Impairment

The diagnosis of AD was based on a neuropsychological examination, cerebrospinal fluid analysis (Aβ, total τ, and phosphorylated τ-protein levels), and brain MRI. For cognitive assessment, we used the Repeatable Battery for the Assessment of Neuropsychological Status (RBANS—population average range 110–119, adjusted to education level, with higher values indicating better performance), in AD patients only: the Mini-Mental State Examination (MMSE) and the Frontal Assessment Battery (FAB). In controls only, we used the Montreal Cognitive Assessment (MoCA) and the Geriatric Depression Scale (GDS). On MRI scans, the Medial Temporal lobe Atrophy scale (MTA, Scheltens) was used to assess cortical atrophy, and the Fazekas scale was used to assess the effect of ischemic white matter changes on cognitive impairment in AD patients.

Controls with normal cognitive performance and the absence of hippocampal atrophy and ischemic white matter lesions on MRI underwent the same test protocol as patients with AD, except for cerebrospinal fluid analysis.

Majority of AD patients were examined early at the time of AD diagnosis (median 7 months; 95% confidence interval 1–14 months).

T2DM patients did not undergo neuropsychological testing, but they did not report any neurological disease in the clinical questionnaire at the time of examination.

### 4.3. Drug Treatment of the Patients

Treatment of AD patients was as follows: 44 patients (50%) received donepezil and rivastigmine, 11 patients (13%) were treated with memantine, and 3 patients (3%) with pi-racetam. Antidepressant drugs were given to 29 AD patients (39%), 22 of them received selective serotonin reuptake inhibitors (citalopram, escitalopram, sertraline), and 7 of them received other antidepressants (trazodone, mirtazapine). In the AD group, 55 patients (63%) had been treated for hypertension, and 23 patients (26%) had dyslipidemia.

T2DM patients were all treated with oral antidiabetics, but none with insulin. In the T2DM group, 58 patients (66%) had been treated for hypertension, and 36 patients (41%) had dyslipidemia.

None of the controls were treated for T2DM or depression, 19 controls (32%) were treated for hypertension, and 18 controls (31%) were taking hypolipidemic drugs.

None of the study participants used hormonal treatment with steroid hormones.

### 4.4. Measurement of Anthropometric Data

Body weight, height, and waist and hip circumferences were measured in order to calculate the body mass index (BMI), body adiposity index (BAI), and the waist-to-hip ratio (WHR). BAI, a surrogate measure of body fat, was calculated as described elsewhere [98].

### 4.5. Biochemical Analyses

For the evaluation of biochemical parameters, blood samples were taken in a fasting state in the morning. Lipid profile assessments included total and high-density lipoprotein (HDL) cholesterol levels, and triacylglycerol concentrations by an enzymatic colorimetric test (Cobas 6000, Roche Diagnostics, Mannheim, Germany). LDL-cholesterol was calculated by the Friedewald–Levy–Fredrickson formula [99].

To assess peripheral insulin sensitivity, the HOMA R was calculated, and for insulin secretion, the homeostasis model of β-cell function (HOMA F) was calculated [100].

For these calculations, blood glucose levels were measured by an enzymatic reference method with hexokinase, insulin, and C-peptide by ECLIA (Cobas 6000, Roche Diagnostics, Mannheim, Germany). Proinsulin levels were measured by ELISA (DRG Proinsulin ELISA, EIA-1560, Marburg, Germany). Uric acid and γ-glutamyl transferase (GGT) levels were measured by an enzymatic colorimetric test (Cobas 6000, Roche Diagnostics, Mannheim, Germany). Alanine aminotransferase (ALT) levels were measured by an IFCC 37 °C method with alanine and pyridoxal phosphate (Roche, Cobas 6000, Roche Diagnostics, Mannheim, Germany), and aspartate aminotransferase (AST) levels were measured by an IFCC 37 °C method with aspartate and pyridoxal phosphate (Roche, Cobas 6000, Roche Diagnostics, Mannheim, Germany). C-reactive protein (CRP) levels were measured by an immunoturbidimetric test (Roche, Cobas 6000, Roche Diagnostics, Mannheim, Germany). Thyrotropin (TSH), free thyroxine (fT4), and free triiodothyronine (fT3) levels were measured by an ECLIA (Roche, Cobas 6000, Roche Diagnostics, Mannheim, Germany). Creatinine levels were measured by an absorption spectrophotometry enzymatic method (Roche, Cobas 6000, Roche Diagnostics, Mannheim, Germany). Free fatty acids (NEFA) levels were measured by an enzymatic colorimetric method (RANDOX Laboratories Limited, Crumlin, UK).

### 4.6. Steroid Analysis

Most steroids and their polar conjugates were measured using our previously described GC-MS method [101]. Here, 17-hydroxy-pregnenolone and its sulfate were analyzed by RIA and RIA after hydrolysis, as reported in our previous papers [102,103]. The 17-hydroxy-progesterone was assayed using a kit from Beckman Coulter, Brea, CA, USA (intra-assay CV = 5.2%, inter-assay CV = 6.5%), cortisol using a RIA kit from Orion, Espoo, Finland (intra-assay CV = 3.8%, inter-assay CV = 4.4%), and SHBG using an IRMA kit from Orion, Espoo, Finland (intra-assay CV = 6.1%, inter-assay CV = 7.9%).

### 4.7. Statistical Analysis

In the first step, the power transformation parameters were found for each metric variable so that its distribution was as close as possible to the Gaussian distribution. The effect of age was assessed using an ANOVA model consisting of the factors AD and T2DM and a T2DM × AD interaction. Considering the significant age differences between groups found by the ANOVA model consisting of the factors AD and T2DM and the interaction AD × T2DM, a two-factor ANCOVA model including the AD and T2DM factors and the AD × T2DM interaction was used, with an adjustment for age for each parameter to determine whether it correlated with AD and/or T2DM. The statistical software Statgraphics Centurion v. XVIII from Statgraphics Technologies, Inc. (The Plains, VA, USA) was used for the above analyses.

However, in terms of interpretation of the results, it was more convenient to use simple OPLS models that examined the correlation of these factors simultaneously with multiple parameters and predicted the presence of AD or T2DM separately for both levels of the remaining factor, or that differentiated between AD patients without T2DM and non-AD patients with T2DM based on multiple parameters. The first model tested correlations between AD and multiple parameters for volunteers without T2DM. The second model tested these correlations for volunteers with T2DM. The third model tested correlations between T2DM and multiple parameters for volunteers without AD, and the fourth model tested these correlations for volunteers with AD. These models were built separately for each sex. The fifth model differentiated AD patients without T2DM and non-AD patients with T2DM based on multiple parameters.

The OPLS model, which is a multivariate regression with dimensionality reduction, allows the evaluation of relationships between explained variables and the explaining variables (predictors) that may be highly correlated, which is also the case for steroids in metabolic pathways. The presence of the observed pathology in the OPLS model is expressed as the logarithm of the likelihood ratio (the ratio of the probability of the presence of pathology, p, to the probability of its absence (1-p)), i.e., the logarithm of the likelihood ratio is calculated, which then ranges from minus infinity to plus infinity. This approach ensures that the prediction of the probability of the presence of pathology is between 0 and 1 (after applying a recurrent formula that converts the prediction of the logarithm of likelihood ratio into a prediction of the probability of the presence of pathology).

The variability in the predictors is divided into two independent components. The first contains the variability of predictors that were shared with the probability of pathology (predictive component), whereas the orthogonal components explained the variability shared within highly correlated predictors. OPLS identifies relevant predictors as well as the best linear combination of predictors to estimate the probability of the presence of pathology. After standardization of the variables, the OPLS model can be expressed as follows:(1)$$X=T_{p}P_{p}^{T}+T_{0}P_{0}^{T}+E$$
(2)$$Y=T_{p}P_{p}^{T}+F$$
where **X** is the matrix with predictors and subjects, **Y** is the vector of the dependent variable and subjects, **T**_p_ is the vector of component scores from the single predictive component and subjects extracted from **Y**, **T**_0_ is the vector of component scores from the single orthogonal component and subjects extracted from **X**, **P**_p_ is the vector of component loadings for the predictive component extracted from **Y**, **P**_0_ is the vector of component loadings for the orthogonal component extracted from **X** and independent variables, and **E** and **F** are the error terms.

The relevant predictors were chosen using variable importance (VIP) statistics. The statistical software SIMCA-P v.12.0 from Umetrics AB (Umeå, Sweden), which was used for OPLS analysis, enabled finding the number of relevant components, the detection of multivariate non-homogeneities, and testing the multivariate normal distribution and homoscedasticity (constant variance).

The algorithm for obtaining the predictions was as follows:Transformation of the original data to obtain the values with symmetric distribution and constant variance.Checking the data homogeneity in predictors using Hotelling’s statistics and the eventual elimination of non-homogeneities.Testing the relevance of predictors using variable importance statistics and the elimination of irrelevant predictors.Calculating component loadings for individual variables to evaluate their correlations with the predictive component.Calculating regression coefficients for the multiple regression model to evaluate the mutual independence of predictors after comparison with the corresponding component loadings from the OPLS model.Calculating predicted values of the logarithm of the ratio of the probability of pathology presence to the probability of pathology absence (LLR).Calculating the probability of the pathology presence for individual subjects.Calculating the sensitivity and specificity of the prediction.

## 5. Conclusions

The most important outcome of this study was the finding that AD (A+D-) and T2DM (A-D+) differed from each other in terms of circulating steroid levels and other monitored characteristics, such as obesity markers, glucose metabolism, liver function tests, and common biochemical markers:In a direct comparison of these two pathologies, excluding patients with comorbidities, AD patients, both men and women, showed significantly lower anthropometric measurements, including body mass index (BMI) and body adiposity index (BAI), compared to T2DM patients.AD patients had a higher insulin secretion index (HOMA F) in comparison with T2DM patients, and women also had a lower insulin resistance index (HOMA R). This probably led to a decrease in circulating glucose levels, which may have contributed to brain malnutrition in AD patients. The direct comparison also found lower uric acid in AD patients compared to T2DM patients, which may also be linked to AD-related brain hypometabolism.Liver function tests mostly inversely correlated with AD regardless of T2DM, supporting the concept of an association between altered liver function on the one hand, and a lower Aβ clearance, increased Aβ deposition, and reduced brain glucose metabolism on the other. This observation was consistent for both sexes.Regarding steroid metabolism, AD patients (both sexes) had, according to the ANCOVA, significantly higher SHBG, cortisol, and 17-hydroxyprogesterone, and lower estradiol and 5α-androstane-3α,17β-diol compared to T2DM patients. Moreover, there were lower progesterone levels in women with AD, and lower allopregnanolone, isopregnanolone, and conjugated 5α-androstane-3β,17β-diol levels in men compared to diabetics. OPLS models included even more relevant predictors and had excellent sensitivity and specificity for both sexes. Sexual dimorphism was seen in two steroid predictors—17-hydroxypregnenolone (positive correlation with AD in women, negative in men) and 5α-dihydrotestosterone (negative correlation with AD in women, positive in men). However, it should be noted that compared to healthy controls, changes in the steroid spectrum (especially increases in levels of steroids from the C21 group, including their 5 α/β-reduced forms, androstenedione, etc.) were similar in patients with AD and patients with T2DM, though more expressed in diabetics. It could be suggested that the involvement of many of these steroids in a contra-regulatory protective mechanism mitigates the development and progression of AD, and T2DM as well.

Surprisingly, the group of patients with comorbidity of AD and T2DM (A+D+) was closer in most screened parameters to patients with AD, or even to controls, and the effect of diabetes was suppressed. These findings require further studies to validate them on a larger cohort.

In conclusion, our results demonstrated the ability of the steroidome to effectively differentiate AD, T2DM, and controls in both men and women, to distinguish the two pathologies from each other, and even to differentiate patients with a combination of AD and T2DM.

## Figures and Tables

**Table 1 ijms-24-08575-t001:** Effects of type 2 diabetes mellitus (T2DM) and Alzheimer’s disease (AD) on anthropometric and biochemical parameters in serum of 136 women, as evaluated by the age-adjusted ANCOVA model. Data are shown as re-transformed means with their 95% confidence intervals: 41 AD women (A+D-), 47 T2DM women (A-D+), 41 control women (A-D-), and 7 women with AD and T2DM (A+D+). The differences for age were evaluated using a two-way ANOVA model.

	T2DM	AD			Factor AD	Factor T2DM	AD× T2DM	Multiple Comparisons
		A-	A+					
Numbers in groups	D-	41	41					
	D+	47	7					
**Anthropometric indices and blood pressure**								
Age (years)	D-	66.4 (64.7, 68.3)	73.7 (71, 76.5)	*F*	**23**	2.8	0.2	A-D- < A+D-, A-D- < A+D+, A-D+ < A+D-, A-D+ < A+D+
	D+	68.4 (66.7, 70.2)	77.7 (71.9, 84.5)	*p*	**0.001**	0.096	0.674	
Abdominal circumference (cm)	D-	95 (91, 98)	92 (88, 97)	*F*	3.8	**4**	1.4	A-D- < A-D+, A-D+ > A+D-
	D+	105 (102, 109)	95 (85, 105)	*p*	0.053	**0.049**	0.234	
Hip circumference (cm)	D-	104 (101, 107)	99.7 (96, 104)	*F*	**8.4**	1.3	2.3	A-D- < A-D+, A-D+ > A+D-, A-D+ > A+D+
	D+	112 (109, 115)	99 (90, 108)	*p*	**0.005**	0.261	0.131	
Waist circumference (cm)	D-	86 (83, 89)	80 (76, 84)	*F*	**8.8**	**7.4**	1.1	A-D- < A-D+, A-D+ > A+D-
	D+	98 (94, 102)	85 (76, 95)	*p*	**0.004**	**0.008**	0.294	
BMI (kg/m^2^)	D-	26.3 (25.1, 27.7)	25.1 (23.6, 26.7)	*F*	**6.8**	**6.1**	2.7	A-D- < A-D+, A-D+ > A+D-, A-D+ > A+D+
	D+	31.3 (29.8, 32.8)	26 (22.9, 29.7)	*p*	**0.01**	**0.015**	0.107	
WHR	D-	0.82 (0.81, 0.84)	0.8 (0.78, 0.82)	*F*	1.2	**11.1**	0.1	A-D- < A-D+, A-D+ > A+D-
	D+	0.88 (0.86, 0.89)	0.86 (0.8, 0.92)	*p*	0.273	**0.001**	0.776	
BAI (%)	D-	32.1 (30.6, 33.7)	30.7 (29, 32.6)	*F*	**10.2**	2.3	**5.6**	A-D- < A-D+, A-D+ > A+D-, A-D+ > A+D+
	D+	37.8 (36.1, 39.5)	29.7 (25.9, 34)	*p*	**0.002**	0.133	**0.02**	
Systolic blood pressure (mmHg)	D-	125 (119, 131)	134 (126, 142)	*F*	0	**4.4**	3.9	A-D- < A-D+
	D+	147 (141, 154)	135 (119, 153)	*p*	0.855	**0.039**	0.051	
Diastolic blood pressure (mmHg)	D-	79 (76, 82)	80 (76, 84)	*F*	1	0.8	1.9	
	D+	80 (77, 83)	74 (63, 82)	*p*	0.313	0.378	0.168	
**Glucose metabolism**								
Glucose (mmol/L)	D-	5.3 (5.2 5.53)	5.1 (4.9, 5.4)	*F*	**14.2**	**17.5**	**7.1**	A-D- < A-D+, A-D+ > A+D-, A-D+ > A+D+
	D+	6.8 (6.5, 7.2)	5.4 (4.9, 5.9)	*p*	**0.001**	**0.001**	**0.009**	
C-peptide (nmol/L)	D-	0.75 (0.67, 0.85)	0.81 (0.69, 0.95)	*F*	0.2	2.9	1.2	A-D- < A-D+
	D+	1.01 (0.9, 1.13)	0.86 (0.62, 1.19)	*p*	0.681	0.091	0.274	
Insulin (mIU/L)	D-	8.3 (7, 9.8)	8.8 (7.1, 11)	*F*	0.2	1.2	1	
	D+	11.1 (9.4, 13.1)	9 (5.8, 14)	*p*	0.635	0.267	0.316	
Proinsulin (pmol/L)	D-	2.73 (2.07, 3.65)	2.37 (1.76, 3.24)	*F*	3.9	1.4	2.1	A-D- < A-D+, A-D+ > A+D-
	D+	4.92 (3.78, 6.5)	2.25 (1.21, 4.47)	*p*	0.052	0.239	0.15	
HOMA R	D-	1.94 (1.65, 2.29)	1.96 (1.58, 2.45)	*F*	2.1	**5.4**	2.7	A-D- < A-D+, A-D+ > A+D-
	D+	3.42 (2.87, 4.12)	2.16 (1.41, 3.4)	*p*	0.155	**0.023**	0.104	
HOMA F	D-	93.3 (77.5, 112)	113 (89.1, 142)	*F*	2.3	1.6	0.1	A-D+ < A+D-
	D+	72.5 (59.8, 87.2)	99 (60.2, 157)	*p*	0.134	0.215	0.712	
**Lipid parameters**								
Total cholesterol (mmol/L)	D-	5.16 (4.91, 5.45)	5.45 (5.09, 5.88)	*F*	**4.8**	0.1	1.3	
	D+	4.99 (4.77, 5.24)	5.87 (5.06, 7.07)	*p*	**0.031**	0.734	0.254	
HDL cholesterol (mmol/L)	D-	1.73 (1.6, 1.87)	1.65 (1.5, 1.82)	*F*	1.2	**7.7**	0.2	A-D- > A-D+
	D+	1.49 (1.39, 1.6)	1.35 (1.1, 1.65)	*p*	0.28	**0.006**	0.664	
LDL cholesterol (mmol/L)	D-	3 (2.73, 3.27)	3.1 (2.76, 3.45)	*F*	**6.6**	1.2	**5.3**	A-D+ < A+D+
	D+	2.74 (2.51, 2.98)	3.93 (3.15, 4.82)	*p*	**0.011**	0.27	**0.023**	
Triacylglycerols (mmol/L)	D-	1 (0.91, 1.12)	1.31 (1.14, 1.53)	*F*	0.3	3	**5.3**	A-D- < A-D+, A-D- < A+D-
	D+	1.47 (1.32, 1.66)	1.25 (0.94, 1.74)	*p*	0.567	0.085	**0.023**	
Free fatty acids (μmol/L)	D-	0.44 (0.39, 0.51)	0.55 (0.46, 0.65)	*F*	**6.9**	3.1	1.1	A-D- < A+D+
	D+	0.48 (0.43, 0.55)	0.75 (0.52, 1.1)	*p*	**0.01**	0.082	0.302	
**Liver function tests**								
Alanine aminotransferase (ALT) (μkat/L)	D-	0.34 (0.3, 0.38)	0.27 (0.24, 0.311)	*F*	**26.4**	0.5	**9.8**	A-D- < A-D+, A-D- > A+D+, A-D+ > A+D-, A-D+ > A+D+
	D+	0.43 (0.38, 0.48)	0.2 (0.16, 0.26)	*p*	**0.001**	0.485	**0.002**	
Aspartate aminotransferase (AST) (μkat/L)	D-	0.38 (0.36, 0.42)	0.35 (0.32, 0.38)	*F*	**7.5**	3.3	1.8	A-D- > A+D+
	D+	0.37 (0.35, 0.4)	0.29 (0.25, 0.35)	*p*	**0.007**	0.073	0.184	
AST/ALT ratio	D-	1.12 (1.06, 1.18)	1.28 (1.2, 1.38)	*F*	**20.9**	1.8	**7.4**	A-D- > A-D+, A-D- < A+D-, A-D- < A+D+, A-D+ < A+D-, A-D+ < A+D+
	D+	0.86 (0.82, 0.91)	1.4 (1.21, 1.62)	*p*	**0.001**	0.188	**0.008**	
γ-Glutamyltransferase (GGT) (μkat/L)	D-	0.33 (0.28, 0.39)	0.35 (0.28, 0.44)	*F*	0.5	0	1.3	
	D+	0.38 (0.32, 0.45)	0.29 (0.19, 0.47)	*p*	0.503	0.906	0.259	
**Thyroid hormones**								
Thyrotropin (TSH) (mIU/L)	D-	2.24 (1.81, 2.72)	1.83 (1.33, 2.4)	*F*	0	1.4	1	
	D+	1.49 (1.17, 1.85)	1.77 (0.86, 2.98)	*p*	0.885	0.243	0.315	
Free thyroxine (fT4) (pmol/L)	D-	15.4 (14.7, 16.1)	14.7 (13.9, 15.5)	*F*	**20.6**	0	**12.8**	A-D- < A-D+, A-D- > A+D+, A-D+ > A+D-, A-D+ > A+D+
	D+	17.8 (17, 18.6)	13.1 (11.8, 14.6)	*p*	**0.001**	0.842	**0.001**	
Free triiodothyronine (fT3) (pmol/L)	D-	4.57 (4.41, 4.74)	4.13 (3.94, 4.32)	*F*	**11.2**	1.8	0	A-D- > A+D-, A-D+ > A+D-
	D+	4.79 (4.63, 4.95)	4.27 (3.88, 4.7)	*p*	**0.001**	0.188	0.844	
**Markers of renal function and C-reactive protein**								
Uric acid (μmol/L)	D-	272 (253, 293)	275 (251, 301)	*F*	**4.7**	0.6	**6.3**	A-D- < A-D+, A-D+ > A+D-, A-D+ > A+D+
	D+	330 (310, 352)	248 (202, 301)	*p*	**0.033**	0.436	**0.013**	
Creatinine (μmol/L)	D-	70.9 (66.6, 75.6)	76.1 (69.9, 83.1)	*F*	0	3.7	1.2	
	D+	67.8 (64, 71.8)	64.7 (55.2, 76.5)	*p*	0.842	0.056	0.269	
C-reactive protein (CRP) (mg/L)	D-	2.1 (1.6, 2.8)	1.4 (1, 1.9)	*F*	0.4	2.7	2.1	
	D+	2.2 (1.7, 2.9)	2.6 (1.3, 5.4)	*p*	0.52	0.107	0.155	

“*F*” is a symbol of the *F*-statistic, “*p*” is the significance level.

**Table 2 ijms-24-08575-t002:** Effects of type 2 diabetes mellitus (T2DM) and Alzheimer’s disease (AD) on anthropometric and biochemical parameters in serum of 83 men, as evaluated by the age-adjusted ANCOVA model. Data are shown as re-transformed means with their 95% confidence intervals: 33 AD men (A+D-), 25 T2DM men (A-D+), 18 control men (A-D-), and 7 men with AD and T2DM (A+D+). The differences for age were evaluated using a two-way ANOVA model.

	T2DM	AD			Factor AD	Factor T2DM	AD× T2DM	Multiple Comparisons
		A-	A+					
Numbers in groups	D-	18	33					
	D+	25	7					
**Anthropometric indices and blood pressure**								
Age (years)	D-	70.7 (68.2, 73.2)	78 (75.8, 80.4)	*F*	**27.7**	0.3	0.4	A-D- < A+D-, A-D- < A+D+, A-D+ < A+D-, A-D+ < A+D+
	D+	69 (66.9, 71.1)	78.1 (73.6, 82.9)	*p*	**0.001**	0.593	0.549	
Abdominal circumference (cm)	D-	100 (96, 105)	97 (94, 100)	*F*	0.6	**7.8**	0.1	A-D+ > A+D-
	D+	108 (103, 113)	106 (97, 118)	*p*	0.447	**0.007**	0.787	
Hip circumference (cm)	D-	102 (99, 105)	100 (98, 102)	*F*	2.8	**9.3**	0.3	A-D- < A-D+, A-D+ > A+D-
	D+	109 (106, 113)	104 (99, 111)	*p*	0.097	**0.003**	0.587	
Waist circumference (cm)	D-	97 (92, 102)	93 (90, 97)	*F*	0.8	**7.1**	0.2	A-D+ > A+D-
	D+	104 (99, 110)	102 (93, 115)	*p*	0.372	**0.01**	0.69	
BMI (kg/m^2^)	D-	27 (25.8, 28.3)	26.3 (25.3, 27.3)	*F*	2.8	3.6	1.1	A-D- < A-D+, A-D+ > A+D-
	D+	30.4 (29, 31.9)	27.1 (25.1, 29.5)	*p*	0.102	0.062	0.307	
WHR	D-	0.95 (0.92, 0.98)	0.92 (0.9, 0.95)	*F*	0.3	1.1	0.6	
	D+	0.95 (0.93, 0.98)	0.96 (0.9, 1.02)	*p*	0.611	0.302	0.431	
BAI (%)	D-	26 (24.7, 27.5)	25.9 (24.8, 27.1)	*F*	2.9	**6.9**	2.6	A-D- < A-D+, A-D+ > A+D-
	D+	30.3 (28.6, 32.3)	26.7 (24.4, 29.7)	*p*	0.092	**0.011**	0.109	
Systolic blood pressure (mmHg)	D-	139 (129, 148)	135 (128, 143)	*F*	1.4	0.1	0.5	
	D+	145 (136, 153)	134 (117, 150)	*p*	0.244	0.718	0.507	
Diastolic blood pressure (mmHg)	D-	81 (77, 85)	80 (77, 84)	*F*	0.6	0.4	0.2	
	D+	81 (77, 85)	78 (71, 85)	*p*	0.427	0.543	0.689	
**Glucose metabolism**								
Glucose (mmol/L)	D-	5.5 (5.2, 5.8)	5.1 (4.9, 5.3)	*F*	**5.3**	**31.6**	0.1	A-D- < A-D+, A-D+ > A+D-, A+D- < A+D+
	D+	6.9 (6.4, 7.5)	6.1 (5.5, 6.9)	*p*	**0.024**	**0.001**	0.749	
C-peptide (nmol/L)	D-	0.78 (0.65, 0.92)	0.98 (0.84, 1.13)	*F*	0.4	0.4	**8.1**	A-D- < A-D+
	D+	1.12 (0.96, 1.32)	0.77 (0.57, 1.05)	*p*	0.534	0.534	**0.006**	
Insulin (mIU/L)	D-	8.4 (6.5, 10.9)	11.7 (9.5, 14.7)	*F*	0.6	0	1.6	
	D+	10.1 (8, 13)	9.4 (6, 15.1)	*p*	0.448	0.924	0.205	
Proinsulin (pmol/L)	D-	1.95 (1.4, 2.76)	2.81 (2.14, 3.72)	*F*	0	**11.9**	3.5	A-D- < A-D+, A-D+ > A+D-
	D+	5.96 (4.32, 8.39)	3.88 (2.14, 7.47)	*p*	0.958	**0.001**	0.065	
HOMA R	D-	2.06 (1.6, 2.67)	2.67 (2.16, 3.32)	*F*	0.1	1.5	2	
	D+	3.13 (2.45, 4.05)	2.59 (1.65, 4.19)	*p*	0.828	0.22	0.164	
HOMA F	D-	91 (68.1, 119)	146 (119, 177)	*F*	3.4	**7.9**	1.1	A-D+ < A+D-
	D+	66.7 (49.2, 87.9)	78.7 (45, 126)	*p*	0.072	**0.007**	0.303	
**Lipid parameters**								
Total cholesterol (mmol/L)	D-	4.99 (4.53, 5.48)	4.85 (4.47, 5.25)	*F*	3.2	**4.1**	2	
	D+	4.83 (4.43, 5.25)	3.93 (3.23, 4.71)	*p*	0.076	**0.047**	0.158	
HDL cholesterol (mmol/L)	D-	1.44 (1.29, 1.58)	1.35 (1.23, 1.47)	*F*	2.3	1.4	0.4	
	D+	1.39 (1.26, 1.51)	1.19 (0.93, 1.45)	*p*	0.135	0.234	0.534	
LDL cholesterol (mmol/L)	D-	2.91 (2.5, 3.36)	2.95 (2.62, 3.31)	*F*	0.9	**5.4**	1.4	
	D+	2.63 (2.3, 2.99)	2.15 (1.58, 2.81)	*p*	0.357	**0.024**	0.249	
Triacylglycerols (mmol/L)	D-	1.07 (0.869, 1.33)	1.07 (0.902, 1.28)	*F*	0.4	2.4	0.4	
	D+	1.42 (1.18, 1.74)	1.2 (0.833, 1.77)	*p*	0.533	0.129	0.513	
Free fatty acids (μmol/L)	D-	0.48 (0.38, 0.59)	0.48 (0.4, 0.56)	*F*	0	1.4	0	
	D+	0.57 (0.47, 0.67)	0.55 (0.36, 0.77)	*p*	0.855	0.24	0.908	
**Liver function tests**								
Alanine aminotransferase (ALT) (μkat/L)	D-	0.42 (0.356, 0.49)	0.31 (0.26, 0.35)	*F*	**5.9**	0.1	0.1	A-D- > A+D-, A-D+ > A+D-
	D+	0.42 (0.36, 0.48)	0.33 (0.22, 0.45)	*p*	**0.018**	0.815	0.726	
Aspartate aminotransferase (AST) (μkat/L)	D-	0.43 (0.39, 0.48)	0.36 (0.34, 0.39)	*F*	**4.7**	0.2	0.5	A-D- > A+D-
	D+	0.43 (0.39, 0.47)	0.39 (0.33, 0.47)	*p*	**0.034**	0.675	p=0.5	
AST/ALT ratio	D-	1.03 (0.92, 1.16)	1.19 (1.07, 1.34)	*F*	**4.1**	0.3	0.1	
	D+	0.97 (0.89, 1.08)	1.16 (0.93, 1.55)	*p*	**0.048**	0.599	0.812	
γ-Glutamyltransferase (GGT) (μkat/L)	D-	0.45 (0.36, 0.57)	0.37 (0.32, 0.44)	*F*	0.2	3.5	0.8	
	D+	0.51 (0.41, 0.64)	0.55 (0.37, 0.89)	*p*	0.666	0.067	0.369	
**Thyroid hormones**								
Thyrotropin (TSH) (mIU/L)	D-	2.03 (1.58, 2.54)	1.77 (1.43, 2.16)	*F*	1.9	0.7	0.3	
	D+	1.94 (1.55, 2.4)	1.43 (0.86, 2.2)	*p*	0.175	0.399	0.584	
Free thyroxine (fT4) (pmol/L)	D-	14.9 (13.9, 15.9)	14.9 (14.1, 15.7)	*F*	0.2	1.7	0.2	
	D+	16 (15.1, 16.9)	15.4 (13.6, 17.3)	*p*	0.654	0.2	0.644	
Free triiodothyronine (fT3) (pmol/L)	D-	4.69 (4.45, 4.95)	4.59 (4.4, 4.8)	*F*	0	0.5	0.3	
	D+	4.48 (4.29, 4.7)	4.56 (4.11, 5.13)	*p*	0.962	0.481	0.591	
**Markers of renal function and C-reactive protein**								
Uric acid (μmol/L)	D-	333 (306, 360)	314 (293, 335)	*F*	**13.9**	1.3	**7.9**	A-D- < A-D+, A-D+ > A+D-, A-D+ > A+D+
	D+	402 (376, 430)	288 (244, 335)	*p*	**0.001**	0.256	**0.007**	
Creatinine (μmol/L)	D-	87.5 (82.4, 93.1)	90.6 (85.9, 95.6)	*F*	2.3	1.6	**4.7**	A-D+ > A+D+, A+D- > A+D+
	D+	92.1 (87.3, 97.3)	74.9 (66.6, 84.8)	*p*	0.137	0.205	**0.033**	
C-reactive protein (CRP) (mg/L)	D-	1.5 (1.1, 2)	1.2 (0.9, 1.5)	*F*	**4.6**	0.1	1.6	A-D+ > A+D-, A-D+ > A+D+
	D+	2.3 (1.7, 3.1)	0.9 (0.6, 1.6)	*p*	**0.035**	0.756	0.208	

“*F*” is a symbol of the *F*-statistic, “*p*” is the significance level.

**Table 3 ijms-24-08575-t003:** Effects of type 2 diabetes mellitus (T2DM) and Alzheimer’s disease (AD) on serum steroids (nmol/L) in 136 women, as evaluated by the age-adjusted ANCOVA model. Data are shown as re-transformed means with their 95% confidence intervals: 41 AD women (A+D-), 47 T2DM women (A-D+), 41 control women (A-D-), and 7 women with AD and T2DM (A+D+). The differences for age were evaluated using a two-way ANOVA model.

	T2DM	AD			Factor AD	Factor T2DM	AD× T2DM	Multiple Comparisons
		A-	A+					
Numbers in groups	D-	41	41					
	D+	47	7					
**SHBG**								
Sex hormone-binding globulin (SHBG)	D-	56.4 (49.5, 63.9)	65.6 (56.5, 75.7)	*F*	0.5	**7.9**	0.6	A-D+ < A+D-
	D+	45 (39.6, 50.8)	45.1 (30.6, 63.1)	*p*	0.477	**0.006**	0.459	
**Δ^5^ C21 steroids**								
Pregnenolone	D-	1.07 (0.876, 1.27)	1.48 (1.2, 1.78)	*F*	0.5	1.6	**9.5**	A-D- < A-D+
	D+	1.89 (1.61, 2.19)	1.15 (0.669, 1.74)	*p*	0.49	0.209	**0.003**	
Pregnenolone sulfate	D-	39.6 (33.6, 46.7)	57.3 (46.8, 70.3)	*F*	3.7	**4.3**	0.3	A-D- < A-D+, A-D- < A+D-
	D+	57.6 (49.6, 67)	71.9 (44.9, 116)	*p*	0.056	**0.04**	0.599	
20α-Dihydropregnenolone	D-	1.23 (1.07, 1.41)	1.56 (1.32, 1.83)	*F*	0	0	1.7	
	D+	1.52 (1.31, 1.76)	1.29 (0.895, 1.82)	*p*	0.833	0.945	0.191	
20α-Dihydropregnenolone sulfate	D-	296 (247, 353)	425 (343, 525)	*F*	0.3	0.2	3.9	A-D- < A-D+
	D+	418 (357, 489)	341 (211, 536)	*p*	0.614	0.676	0.051	
17-Hydroxypregnenolone	D-	0.939 (0.569, 1.51)	4.69 (2.67, 8.16)	*F*	3.2	0	**5.9**	A-D- < A-D+, A-D- < A+D-
	D+	2.26 (1.55, 3.28)	1.86 (0.529, 5.94)	*p*	0.079	0.913	**0.017**	
16α-Hydroxypregnenolone	D-	0.169 (0.137, 0.205)	0.288 (0.229, 0.358)	*F*	0	0.2	**12.3**	A-D- < A-D+, A-D- < A+D-
	D+	0.273 (0.221, 0.334)	0.152 (0.0838, 0.253)	*p*	0.9	0.63	**0.001**	
**Δ^5^ C19 steroids**								
Dehydroepiandrosterone	D-	7.06 (5.63, 8.77)	11.1 (8.64, 14.1)	*F*	0.7	3.4	3.6	
	D+	7.15 (5.6, 9.03)	6 (3.3, 10.2)	*p*	0.415	0.067	0.061	
Dehydroepiandrosterone sulfate	D-	680 (546, 837)	769 (590, 988)	*F*	0	1.1	0.7	
	D+	650 (531, 787)	549 (288, 955)	*p*	0.921	0.288	0.411	
7α-Hydroxy-dehydroepiandrosterone	D-	0.412 (0.334, 0.501)	0.566 (0.448, 0.706)	*F*	1.3	0.8	**10.2**	
	D+	0.596 (0.482, 0.728)	0.284 (0.143, 0.49)	*p*	0.267	0.389	**0.002**	
Conjugated 16α-hydroxy-dehydroepiandrosterone	D-	1.46 (1.08, 2)	2.37 (1.6, 3.57)	*F*	0.4	0.1	**6.6**	
	D+	2.66 (1.98, 3.6)	1.15 (0.504, 2.62)	*p*	0.524	0.798	**0.012**	
Androstenediol	D-	0.802 (0.667, 0.954)	0.927 (0.742, 1.14)	*F*	1.6	2.9	**5.3**	
	D+	0.876 (0.716, 1.06)	0.471 (0.233, 0.82)	*p*	0.211	0.091	**0.023**	
Androstenediol sulfate	D-	307 (239, 397)	295 (217, 405)	*F*	**5.3**	0.1	**5.3**	A-D+ > A+D+
	D+	475 (373, 612)	177 (92.2, 339)	*p*	**0.023**	0.808	**0.024**	
5-Androstene-3β,7α,17β-triol	D-	0.0967 (0.0773, 0.12)	0.0901 (0.068, 0.118)	*F*	3.7	0	2.8	
	D+	0.135 (0.107, 0.171)	0.0682 (0.0348, 0.122)	*p*	0.059	0.833	0.095	
5-Androstene-3β,7β,17β-triol	D-	0.0702 (0.0582, 0.0844)	0.0643 (0.0509, 0.0807)	*F*	**8.1**	0.4	**6.3**	A-D+ > A+D+
	D+	0.0935 (0.0761, 0.115)	0.0386 (0.0215, 0.0644)	*p*	**0.006**	0.539	**0.014**	
**Progestogens**								
Progesterone	D-	0.526 (0.427, 0.639)	0.53 (0.41, 0.673)	*F*	**4.4**	1.7	**5.4**	A-D- < A-D+, A-D+ > A+D-, A-D+ > A+D+
	D+	0.927 (0.754, 1.13)	0.447 (0.241, 0.747)	*p*	**0.038**	0.191	**0.022**	
17-Hydroxyprogesterone	D-	1.16 (0.983, 1.38)	1.9 (1.51, 2.45)	*F*	**9.6**	1.8	**0**	A-D- < A+D-, A-D+ < A+D-
	D+	0.969 (0.836, 1.13)	1.55 (0.99, 2.6)	*p*	**0.002**	0.187	0.998	
**Cortisol**								
Cortisol	D-	473 (430, 522)	647 (573, 732)	*F*	**6.7**	0.3	1.3	A-D- < A+D-, A-D+ < A+D-
	D+	495 (453, 540)	564 (438, 730)	*p*	**0.011**	0.569	0.258	
**Androstenedione and active androgens**								
Androstenedione	D-	1.74 (1.43, 2.11)	2.82 (2.23, 3.56)	*F*	0.2	2.5	**6.7**	A-D- < A-D+, A-D- < A+D-
	D+	3.37 (2.72, 4.19)	2.42 (1.46, 3.98)	*p*	0.661	0.116	**0.011**	
Testosterone	D-	0.935 (0.756, 1.16)	0.822 (0.633, 1.07)	*F*	1.7	0.8	0.5	
	D+	1.24 (0.974, 1.58)	0.86 (0.49, 1.51)	*p*	0.201	0.364	0.506	
5α-Dihydrotestosterone	D-	0.206 (0.171, 0.246)	0.247 (0.195, 0.31)	*F*	1.2	**4.3**	0	
	D+	0.283 (0.237, 0.337)	0.33 (0.209, 0.503)	*p*	0.276	**0.042**	0.948	
**Estrogens**								
Estrone	D-	0.138 (0.116, 0.164)	0.144 (0.117, 0.18)	*F*	2.4	0.5	3.9	A-D- < A-D+
	D+	0.208 (0.168, 0.259)	0.121 (0.0776, 0.192)	*p*	0.127	0.47	0.05	
Estradiol	D-	0.0455 (0.0374, 0.0557)	0.061 (0.0472, 0.0801)	*F*	2.7	**7.8**	**12.2**	A-D- < A-D+, A-D+ > A+D-, A-D+ > A+D+
	D+	0.156 (0.114, 0.224)	0.0545 (0.0327, 0.0974)	*p*	0.106	**0.006**	**0.001**	
**5α/β-Reduced C21 steroids**								
Allopregnanolone	D-	0.11 (0.092, 0.133)	0.14 (0.112, 0.175)	*F*	0	0.1	2.2	
	D+	0.143 (0.117, 0.176)	0.116 (0.0717, 0.186)	*p*	0.949	0.813	0.139	
Allopregnanolone sulfate	D-	2.17 (1.84, 2.56)	2.53 (2.07, 3.11)	*F*	0.6	**6**	0.1	A-D- < A-D+
	D+	3.17 (2.72, 3.69)	3.41 (2.2, 5.4)	*p*	0.435	**0.016**	0.755	
Isopregnanolone	D-	0.0897 (0.075, 0.107)	0.139 (0.112, 0.174)	*F*	3.1	0	**23.6**	A-D- < A-D+, A-D- < A+D-, A-D+ > A+D+, A+D- > A+D+
	D+	0.181 (0.147, 0.223)	0.0659 (0.0405, 0.105)	*p*	0.084	0.864	**0.001**	
Isopregnanolone sulfate	D-	3.25 (2.78, 3.81)	3.82 (3.14, 4.68)	*F*	1.3	0.7	**6**	A-D- < A-D+
	D+	5.02 (4.33, 5.85)	3.11 (2.06, 4.73)	*p*	0.266	0.403	**0.016**	
Pregnanolone	D-	0.0609 (0.048, 0.0783)	0.0855 (0.0614, 0.123)	*F*	0	0	2.7	
	D+	0.0866 (0.0677, 0.113)	0.0611 (0.0333, 0.125)	*p*	0.984	0.969	0.108	
Conjugated pregnanolone	D-	7.18 (6.05, 8.48)	11.7 (9.65, 14.2)	*F*	0.2	1.2	**10.5**	A-D- < A-D+, A-D- < A+D-
	D+	12.7 (11, 14.6)	8.88 (5.69, 13.4)	*p*	0.667	0.28	**0.002**	
Conjugated 5α-pregnane-3β,20α-diol	D-	132 (104, 168)	182 (135, 246)	*F*	0.4	0.4	1	
	D+	141 (113, 176)	132 (70.2, 251)	*p*	0.546	0.535	0.333	
Conjugated 5β-pregnane-3α,20α-diol	D-	9.01 (7.54, 10.7)	15.6 (13, 18.7)	*F*	3.6	0	**4.7**	A-D- < A+D-
	D+	12 (10.4, 13.8)	11.8 (7.52, 17.5)	*p*	0.062	0.95	**0.032**	
**5α/β-Reduced C19 steroids**								
Androsterone	D-	0.221 (0.184, 0.265)	0.31 (0.249, 0.387)	*F*	0	0	**4.5**	
	D+	0.311 (0.254, 0.381)	0.231 (0.143, 0.371)	*p*	0.892	0.877	**0.037**	
Androsterone sulfate	D-	331 (256, 425)	311 (225, 423)	*F*	1.6	1	1.1	
	D+	333 (264, 417)	198 (91.1, 391)	*p*	0.207	0.31	0.293	
Epiandrosterone	D-	0.408 (0.346, 0.479)	0.562 (0.462, 0.683)	*F*	0	3.6	**5.9**	
	D+	0.436 (0.362, 0.524)	0.312 (0.196, 0.479)	*p*	0.99	0.06	**0.017**	
Epiandrosterone sulfate	D-	120 (95.9, 148)	113 (85.6, 148)	*F*	2	0.7	1.5	
	D+	128 (105, 156)	77 (39.4, 139)	*p*	0.156	0.41	0.225	
Etiocholanolone	D-	0.153 (0.122, 0.195)	0.235 (0.172, 0.331)	*F*	0.1	**4.3**	2.8	
	D+	0.141 (0.109, 0.185)	0.11 (0.0615, 0.205)	*p*	0.721	**0.041**	0.099	
Etiocholanolone sulfate	D-	27.2 (21.8, 34.1)	39.8 (30.1, 53)	*F*	0	3.7	**4.3**	
	D+	27.9 (22.8, 34.2)	19.1 (10.4, 34.3)	*p*	0.995	0.058	**0.041**	
Epietiocholaniolone sulfate	D-	14.8 (10.9, 20.1)	18.8 (12.8, 27.5)	*F*	1	1.4	**4.2**	
	D+	18.3 (13.8, 24.2)	8.28 (3.47, 18.6)	*p*	0.313	0.239	**0.044**	
5α-Androstane-3α,17β-diol	D-	0.0424 (0.0289, 0.0608)	0.0642 (0.0412, 0.0974)	*F*	2.6	2.9	**9.8**	A-D- < A-D+, A-D+ > A+D-, A-D+ > A+D+
	D+	0.165 (0.115, 0.234)	0.042 (0.0143, 0.104)	*p*	0.111	0.094	**0.002**	
Conjugated 5α-androstane-3α,17β-diol	D-	10.3 (7.08, 14.5)	13.6 (8.83, 20.2)	*F*	0.7	0.4	3.6	A-D- < A-D+
	D+	20.3 (15.3, 26.6)	9.45 (3.03, 22.3)	*p*	396	0.542	0.059	
Conjugated 5α-androstane-3β,17β-diol	D-	34.4 (26.6, 44.5)	27.4 (19.9, 37.6)	*F*	**8.2**	0.2	3.9	A-D+ > A+D-, A-D+ > A+D+
	D+	47.8 (37.8, 60.6)	16.6 (8.41, 32.2)	*p*	**0.005**	0.677	0.05	
Conjugated 5β-androstane-3α,17β-diol	D-	5.21 (4.17, 6.56)	5.69 (4.3, 7.6)	*F*	3.4	2.1	**6**	A-D+ > A+D+
	D+	6.27 (5.09, 7.77)	2.79 (1.55, 4.98)	*p*	0.069	0.15	**0.016**	

“*F*” is a symbol of the *F*-statistic, “*p*” is the significance level.

**Table 4 ijms-24-08575-t004:** Effects of type 2 diabetes mellitus (T2DM) and Alzheimer’s disease (AD) on serum steroids (nmol/L) in 83 men, as evaluated by the age-adjusted ANCOVA model. Data are shown as re-transformed means with their 95% confidence intervals: 33 AD men (A+D-), 25 T2DM men (A-D+), 18 control men (A-D-), and 7 men with AD and T2DM (A+D+). The differences for age were evaluated using a two-way ANOVA model.

Substance	T2DM	AD			Factor AD	Factor T2DM	AD× T2DM	Multiple Comparisons
		A-	A+					
Numbers in groups	D-	18	33					
	D+	25	7					
**SHBG**								
Sex hormone-binding globulin (SHBG)	D-	40 (33.9, 47.2)	52.6 (45.9, 60.4)	*F*	**5.1**	0.4	0.1	A-D+ < A+D-
	D+	38.2 (33.2, 44)	48 (35.1, 66.3)	*p*	**0.027**	0.511	0.828	
**Δ^5^ C21 steroids**								
Pregnenolone	D-	1.35 (1.08, 1.69)	1.61 (1.34, 1.95)	*F*	0.6	0.1	**4.6**	
	D+	1.74 (1.39, 2.22)	1.16 (0.799, 1.71)	*p*	0.432	0.781	**0.037**	
Pregnenolone sulfate	D-	99.5 (77.8, 127)	112 (91.2, 137)	*F*	0.2	0.7	1.4	
	D+	105 (84.5, 130)	82.3 (52.6, 127)	*p*	0.694	0.394	0.235	
20α-Dihydropregnenolone	D-	1.4 (1.13, 1.75)	1.46 (1.23, 1.75)	*F*	0.6	0.8	1.3	
	D+	1.44 (1.17, 1.8)	1.12 (0.795, 1.63)	*p*	0.453	0.365	0.265	
20α-Dihydropregnenolone sulfate	D-	1010 (759, 1360)	1070 (845, 1370)	*F*	0.2	0.5	**0**	
	D+	881 (688, 1130)	971 (590, 1650)	*p*	0.676	0.504	0.91	
17-Hydroxypregnenolone	D-	2.21 (1.25, 3.83)	1.89 (1.16, 3.01)	*F*	0.4	**4.7**	1.2	
	D+	3.22 (2.06, 5.03)	5.97 (2.02, 18.3)	*p*	0.532	**0.034**	0.279	
16α-Hydroxypregnenolone	D-	0.296 (0.235, 0.371)	0.339 (0.282, 0.407)	*F*	0.4	0.3	2.7	
	D+	0.345 (0.276, 0.432)	0.252 (0.166, 0.375)	*p*	0.547	0.613	0.109	
**Δ^5^ C19 steroids**								
Dehydroepiandrosterone	D-	7.59 (5.86, 9.85)	8.28 (6.71, 10.2)	*F*	0.2	0.1	0	
	D+	8.05 (6.23, 10.4)	8.49 (5.39, 13.5)	*p*	0.68	0.794	0.915	
Dehydroepiandrosterone sulfate	D-	1450 (1120, 1910)	1310 (1060, 1630)	*F*	0	2.5	0.4	
	D+	908 (723, 1140)	1070 (677, 1710)	*p*	0.891	0.12	0.534	
7α-Hydroxy-dehydroepiandrosterone	D-	0.501 (0.414, 0.606)	0.531 (0.455, 0.619)	*F*	0.5	0.2	1.3	
	D+	0.64 (0.531, 0.773)	0.473 (0.337, 0.659)	*p*	0.469	0.683	0.256	
Conjugated 16α-hydroxy-dehydroepiandrosterone	D-	5.09 (2.89, 8.56)	6.87 (4.43, 10.4)	*F*	2.2	0.3	0.4	
	D+	4.84 (2.95, 7.68)	9.73 (3.85, 22.4)	*p*	0.144	0.619	0.52	
Androstenediol	D-	1.8 (1.47, 2.21)	1.66 (1.41, 1.96)	*F*	3.9	2.2	2.1	
	D+	1.8 (1.47, 2.2)	1.15 (0.785, 1.65)	*p*	0.054	0.143	0.149	
Androstenediol sulfate	D-	1140 (770, 1730)	787 (572, 1090)	*F*	0.8	0	0.4	
	D+	968 (687, 1380)	897 (452, 1850)	*p*	0.384	0.945	0.538	
5-Androstene-3β,7α,17β-triol	D-	0.151 (0.114, 0.2)	0.14 (0.112, 0.176)	*F*	0.7	0.1	0.2	
	D+	0.154 (0.117, 0.205)	0.124 (0.0765, 0.203)	*p*	0.418	0.77	0.668	
5-Androstene-3β,7β,17β-triol	D-	0.0947 (0.0717, 0.125)	0.0929 (0.0741, 0.117)	*F*	0.1	0.2	0.1	
	D+	0.0918 (0.0696, 0.121)	0.0837 (0.051, 0.137)	*p*	0.761	0.688	0.832	
**Progestogens**								
Progesterone	D-	0.605 (0.456, 0.788)	0.679 (0.545, 0.838)	*F*	3.5	0.3	**7.3**	A-D+ > A+D+
	D+	0.857 (0.663, 1.09)	0.386 (0.21, 0.639)	*p*	0.066	0.595	**0.009**	
17-Hydroxyprogesterone	D-	3.23 (2.63, 3.93)	3.76 (3.18, 4.42)	*F*	2.4	**8.2**	0.2	A-D- > A-D+, A-D+ < A+D-
	D+	2.13 (1.75, 2.58)	2.79 (1.9, 3.98)	*p*	0.125	**0.006**	0.687	
**Cortisol**								
Cortisol	D-	470 (420, 529)	605 (545, 674)	*F*	2.8	0.5	2.7	A-D- < A+D-
	D+	502 (453, 558)	507 (414, 631)	*p*	0.099	0.464	0.105	
**Androstenedione and active androgens**								
Androstenedione	D-	3.05 (2.42, 3.86)	3.49 (2.89, 4.24)	*F*	0.5	1.5	3	
	D+	4.67 (3.68, 6)	3.24 (2.16, 4.93)	*p*	0.469	0.233	0.09	
Testosterone	D-	14.3 (11.7, 17.3)	14 (11.9, 16.5)	*F*	2	0.1	2	
	D+	17.2 (14.3, 20.8)	12.2 (8.51, 17.1)	*p*	0.158	0.827	0.166	
5α-Dihydrotestosterone	D-	1.55 (1.11, 2.18)	1.34 (1.02, 1.76)	*F*	0	2.6	0.4	
	D+	0.987 (0.701, 1.38)	1.09 (0.594, 1.97)	*p*	0.914	0.113	0.55	
**Estrogens**								
Estrone	D-	0.225 (0.188, 0.272)	0.19 (0.166, 0.219)	*F*	**4**	0	0.3	
	D+	0.242 (0.203, 0.292)	0.181 (0.137, 0.246)	*p*	**0.049**	0.92	0.585	
Estradiol	D-	0.127 (0.105, 0.156)	0.111 (0.0962, 0.13)	*F*	**5.2**	**10.1**	1.8	A-D- < A-D+, A-D+ > A+D-
	D+	0.243 (0.186, 0.34)	0.138 (0.0991, 0.207)	*p*	**0.025**	**0.002**	0.182	
**5α/β-Reduced C21 steroids**								
Allopregnanolone	D-	0.14 (0.112, 0.174)	0.128 (0.107, 0.152)	*F*	**4.8**	0.8	2.8	A-D+ > A+D-
	D+	0.194 (0.157, 0.239)	0.115 (0.0778, 0.167)	*p*	**0.032**	0.389	0.099	
Allopregnanolone sulfate	D-	4.27 (3.28, 5.52)	3.92 (3.15, 4.85)	*F*	3.4	0.2	2	
	D+	5.66 (4.53, 7.05)	3.32 (2.01, 5.26)	*p*	0.07	0.693	0.157	
Isopregnanolone	D-	0.0886 (0.0708, 0.112)	0.107 (0.0882, 0.132)	*F*	1.8	2.1	**7.2**	A-D- < A-D+, A-D+ > A+D-
	D+	0.172 (0.132, 0.228)	0.0889 (0.0583, 0.142)	*p*	0.19	0.154	**0.009**	
Isopregnanolone sulfate	D-	6.2 (4.89, 7.79)	6.69 (5.53, 8.07)	*F*	3.8	0.2	**7**	A-D- < A-D+, A-D+ > A+D+
	D+	9.51 (7.82, 11.6)	4.92 (3.13, 7.42)	*p*	0.054	0.63	**0.01**	
Pregnanolone	D-	0.0665 (0.0454, 0.0978)	0.119 (0.0863, 0.165)	*F*	3.3	0.2	0.2	
	D+	0.067 (0.0459, 0.0981)	0.0946 (0.0483, 0.191)	*p*	0.073	0.648	0.627	
Conjugated pregnanolone	D-	16.7 (13.3, 20.7)	22.1 (18.6, 26.3)	*F*	0.7	0.7	**9.3**	A-D- < A-D+
	D+	27.4 (22.8, 33)	16.5 (11, 24.1)	*p*	0.41	0.421	**0.003**	
Conjugated 5α-pregnane-3β,20α-diol	D-	220 (163, 294)	197 (154, 251)	*F*	0.2	0	1.3	
	D+	185 (142, 239)	248 (147, 412)	*p*	0.63	0.872	0.262	
Conjugated 5β-pregnane-3α,20α-diol	D-	13.5 (10.1, 17.8)	23.5 (18.8, 29.3)	*F*	3.9	0.1	1.5	A-D- < A+D-
	D+	15.9 (12.4, 20.2)	18.4 (11.3, 29.7)	*p*	0.052	0.798	0.225	
**5α/β-Reduced C19 steroids**								
Androsterone	D-	0.496 (0.404, 0.597)	0.418 (0.348, 0.495)	*F*	**5.5**	0.2	1.3	
	D+	0.589 (0.49, 0.698)	0.378 (0.244, 0.54)	*p*	**0.022**	0.704	0.26	
Androsterone sulfate	D-	689 (438, 1060)	556 (381, 801)	*F*	1.1	2	0.1	
	D+	516 (343, 763)	344 (138, 767)	*p*	0.3	0.165	0.743	
Epiandrosterone	D-	0.433 (0.345, 0.541)	0.461 (0.384, 0.552)	*F*	0.1	0	0.7	
	D+	0.491 (0.394, 0.611)	0.416 (0.277, 0.614)	*p*	0.727	0.931	0.406	
Epiandrosterone sulfate	D-	307 (205, 458)	237 (169, 330)	*F*	0.8	2.1	0	
	D+	209 (146, 298)	171 (79.7, 349)	*p*	0.385	0.149	0.9	
Etiocholanolone	D-	0.141 (0.112, 0.178)	0.196 (0.162, 0.238)	*F*	0.6	0.7	2.2	
	D+	0.155 (0.123, 0.195)	0.14 (0.0937, 0.211)	*p*	0.455	0.395	0.141	
Etiocholanolone sulfate	D-	29.6 (21.6, 40.6)	45.3 (34.9, 59)	*F*	0	0.2	**4.5**	
	D+	40.9 (31, 54.1)	27.8 (15.7, 48.6)	*p*	0.924	0.667	**0.038**	
Epietiocholanolone sulfate	D-	25.6 (16.2, 41.2)	24.4 (16.8, 36)	*F*	1.1	0.5	0.9	
	D+	27.2 (18.2, 41.3)	15.5 (6.78, 35)	*p*	0.309	0.471	0.355	
5α-Androstane-3α,17β-diol	D-	0.168 (0.107, 0.246)	0.141 (0.0948, 0.198)	*F*	**8.9**	**4.9**	**6.4**	A-D- < A-D+, A-D+ > A+D-, A-D+ > A+D+
	D+	0.449 (0.333, 0.588)	0.129 (0.0488, 0.258)	*p*	**0.004**	**0.03**	**0.014**	
Conjugated 5α-androstane-3α,17β-diol	D-	98.3 (53.3, 171)	48.3 (27, 81.1)	*F*	**4.6**	0.5	0.2	A-D+ > A+D-
	D+	142 (86.5, 226)	54.6 (14.3, 154)	*p*	**0.036**	0.484	0.697	
Conjugated 5α-androstane-3β,17β-diol	D-	194 (126, 294)	94.3 (64, 136)	*F*	3.9	0.7	0.4	A-D+ > A+D-
	D+	205 (141, 294)	139 (61.6, 291)	*p*	0.052	0.412	0.543	
Conjugated 5β-androstane-3α,17β-diol	D-	12.8 (9.2, 17.7)	13.9 (10.7, 18.2)	*F*	0.5	2.8	1.4	
	D+	22.4 (16.8, 29.9)	15.3 (8.57, 27.2)	*p*	0.487	0.099	0.237	

“*F*” is a symbol of the *F*-statistic, “*p*” is the significance level.

**Table 5 ijms-24-08575-t005:** Relationships between 41 women with AD (A+D-) vs. 41 control women (A-D-). Logarithm of likelihood ratio (LLR) and relevant predictors, as evaluated by the OPLS model (for details, see the Section 4.7).

	Variable	Component Loading	t-Statistic	R	
Relevant predictors (matrix **X**)	Age	0.244	6.63	0.569	**
	Waist circumference	−0.149	−1.69	−0.348	
	WHR	−0.138	−1.63	−0.322	
	Systolic blood pressure	0.149	2.59	0.347	**
	Glucose	−0.131	−1.93	−0.304	**
	Triacylglycerols	0.152	1.77	0.355	
	Creatinine	0.145	2.47	0.357	**
	Alanine aminotransferase (ALT)	−0.280	−4.99	−0.653	**
	Aspartate aminotransferase (AST)	−0.123	−2.90	−0.286	**
	AST/ALT ratio	0.281	4.28	0.655	**
	Free triiodothyronine (fT3)	−0.252	−3.99	−0.586	**
	Pregnenolone	0.182	4.31	0.424	**
	Pregnenolone sulfate	0.174	2.05	0.405	**
	17-Hydroxypregnenolone	0.238	7.02	0.579	**
	16α-Hydroxypregnenolone	0.189	12.04	0.440	**
	20α-Dihydropregnenolone	0.106	2.94	0.247	**
	20α-Dihydropregnenolone sulfate	0.164	3.42	0.382	**
	Dehydroepiandrosterone (DHEA)	0.117	4.42	0.274	**
	7α-Hydroxy-DHEA	0.102	3.82	0.238	**
	16α-Hydroxy-DHEA sulfate	0.182	3.25	0.425	**
	17-Hydroxyprogesterone	0.236	5.28	0.545	**
	Cortisol	0.257	7.67	0.594	**
	Androstenedione	0.071	1.10	0.164	
	Isopregnanolone	0.140	5.32	0.326	**
	Pregnanolone	0.155	3.55	0.389	**
	Conjugated pregnanolone	0.200	2.86	0.466	**
	Conjugated 5β-pregnane−3α,20α-diol	0.251	3.95	0.584	**
	Androsterone	0.102	2.52	0.237	**
	Androsterone sulfate	−0.122	−1.70	−0.284	
	Etiocholanolone	0.062	1.64	0.144	
	Sex hormone-binding globulin (SHBG)	0.205	2.61	0.474	**
	A+D- vs. A-D- group (LLR)	1.000	17.45	0.782	**
**Explained variability**		61.2% (53.4% after cross-validation)			
Sensitivity = 0.923 (0.821, 1.026), Specificity = 0.902 (0.812, 0.993)					

R: Component loadings expressed as correlation coefficients with a predictive component, ** *p* < 0.01.

**Table 6 ijms-24-08575-t006:** Relationships between 33 men with AD (A+D-) vs. 18 control men (A-D-). Logarithm of likelihood ratio (LLR) and relevant predictors, as evaluated by the OPLS model (for details, see the Section 4.7).

	Variable	Component Loading	t-Statistic	R	
Relevant predictors (matrix **X**)	Age	0.279	4.63	0.545	**
	Glucose	−0.275	−2.36	−0.537	*
	C-peptide	0.067	0.60	0.131	
	Proinsulin	0.184	1.49	0.360	
	HOMA F	0.249	1.30	0.487	
	Alanine aminotransferase (ALT)	−0.380	−3.46	−0.742	**
	Aspartate aminotransferase (AST)	−0.199	−3.18	−0.388	**
	AST/ALT ratio	0.331	3.06	0.648	**
	Sex hormone-binding globulin (SHBG)	0.269	3.25	0.529	**
	Cortisol (RIA)	0.253	2.61	0.497	*
	Isopregnanolone	0.300	2.76	0.586	*
	Conjugated 5β-pregnane-3α,20α-diol	0.310	5.73	0.606	**
	Etiocholanolone	0.309	5.36	0.603	**
	Etiocholanolone sulfate	0.198	2.33	0.387	*
	Conjugated 5α-androstane-3β,17β-diol	−0.155	−1.80	−0.302	
	A+D- vs. A-D- group (LLR)	1.000	12.55	0.771	**
**Explained variability**		59.4% (56.5% after cross-validation)			
Sensitivity = 0.923 (0.821, 1.026), Specificity = 0.833 (0.661, 1.006)					

R: Component loadings expressed as correlation coefficients with a predictive component, * *p* < 0.05, ** *p* < 0.01.

**Table 7 ijms-24-08575-t007:** Relationships between 47 women with T2DM (A-D+) vs. 41 control women (A-D-). Logarithm of likelihood ratio (LLR) and relevant predictors, as evaluated by the OPLS model (for details, see the Section 4.7).

	Variable	Component Loading	t-Statistic	R	
Relevant predictors (matrix **X**)	Abdominal circumference	0.188	4.23	0.491	**
	Hip circumference	0.162	5.98	0.428	**
	Waist circumference	0.201	5.74	0.528	**
	BMI	0.214	6.18	0.568	**
	WHR	0.158	3.17	0.411	**
	BAI	0.217	6.32	0.579	**
	Systolic blood pressure	0.209	8.19	0.555	**
	Glucose	0.286	6.76	0.768	**
	C-peptide	0.117	3.73	0.282	**
	Insulin	0.090	2.60	0.275	*
	Proinsulin	0.185	5.24	0.467	**
	HOMA R	0.161	6.79	0.460	**
	HDL cholesterol	−0.118	−3.42	−0.302	**
	Triacylglycerols	0.198	3.74	0.515	**
	Alanine aminotransferase (ALT)	0.091	2.33	0.226	*
	AST/ALT ratio	−0.146	−3.25	−0.371	**
	Free thyroxine (fT4)	0.191	3.64	0.505	**
	Uric acid (UA)	0.162	3.53	0.419	**
	Sex hormone-binding globulin (SHBG)	−0.094	−3.32	−0.239	**
	Pregnenolone	0.190	7.23	0.449	**
	Pregnenolone sulfate	0.124	2.24	0.320	*
	17-Hydroxypregnenolone	0.081	3.90	0.260	**
	16α-Hydroxypregnenolone	0.140	11.88	0.342	**
	20α-Dihydropregnenolone sulfate	0.100	2.02	0.267	*
	16α-Hydroxy-DHEA sulfate	0.104	3.00	0.286	*
	Progesterone	0.163	3.71	0.424	**
	Androstenedione	0.201	5.52	0.516	**
	5α-Dihydrotestosterone	0.132	3.52	0.390	**
	Estrone	0.142	2.92	0.358	*
	Estradiol	0.223	3.77	0.582	**
	Allopregnanolone sulfate	0.103	1.39	0.259	
	Isopregnanolone	0.205	7.00	0.505	**
	Isopregnanolone sulfate	0.131	2.22	0.339	*
	Pregnanolone	0.096	3.77	0.311	**
	Conjugated pregnanolone	0.202	4.94	0.528	**
	Conjugated 5β-pregnane-3α,20α-diol	0.108	2.03	0.294	*
	5α-Androstane-3α,17β-diol	0.201	5.83	0.528	**
	A-D+ vs. A-D- group (LLR)	1.000	13.97	0.903	**
**Explained variability**		81.4% (73.3% after cross-validation)			
Sensitivity = 1 (1, 1), Specificity = 0.941 (0.829, 1.053)					

R: Component loadings expressed as correlation coefficients with a predictive component, * *p* < 0.05, ** *p* < 0.01.

**Table 8 ijms-24-08575-t008:** Relationships between 25 men with T2DM (A-D+) vs. 18 control men (A-D-). Logarithm of likelihood ratio (LLR) and relevant predictors, as evaluated by the OPLS model (for details, see the Section 4.7).

	Variable	Component Loading	t-Statistic	R	
Relevant predictors (matrix **X**)	Abdominal circumference	0.117	2.32	0.258	*
	Hip circumference	0.183	3.55	0.412	**
	Waist circumference	0.122	2.13	0.267	*
	BMI	0.163	3.54	0.385	**
	BAI	0.184	3.12	0.408	**
	Glucose	0.242	3.72	0.551	**
	C-peptide	0.213	4.99	0.485	**
	Proinsulin	0.300	8.48	0.685	**
	HOMA R	0.205	3.92	0.456	**
	Triacylglycerols	0.107	2.59	0.234	*
	AST/ALT ratio	−0.138	−2.13	−0.311	*
	17-Hydroxypregnenolone	−0.162	−6.70	−0.363	**
	Progesterone	0.228	3.85	0.550	**
	Androstenedione	0.246	3.00	0.588	**
	5α-Dihydrotestosterone	−0.105	−1.71	−0.249	
	Estradiol	0.290	20.79	0.695	**
	Isopregnanolone	0.282	4.54	0.681	**
	Isopregnanolone sulfate	0.258	5.62	0.597	**
	Conjugated pregnanolone	0.308	4.63	0.713	**
	5α-Androstane-3α,17β-diol	0.289	5.11	0.686	**
	Conjugated 5β-androstane-3α,17β-diol	0.278	4.88	0.639	**
	A-D+ vs. A-D- group (LLR)	1.000	16.78	0.871	**
**Explained variability**		75.9% (69.1% after cross-validation)			
Sensitivity = 1 (1, 1), Specificity = 0.941 (0.829, 1.053)					

R: Component loadings expressed as correlation coefficients with a predictive component, * *p* < 0.05, ** *p* < 0.01.

**Table 9 ijms-24-08575-t009:** Relationships between 41 women with AD (A+D-) vs. 47 women with T2DM (A-D+). Logarithm of likelihood ratio (LLR) and relevant predictors, as evaluated by the OPLS model (for details, see the Section 4.7).

	Variable	Component Loading	t-Statistic	R	
Relevant predictors (matrix **X**)	Age	0.125	3.46	0.343	**
	Abdominal circumference	−0.187	−7.78	−0.525	**
	Hip circumference	−0.203	−7.08	−0.567	**
	Waist circumference	−0.254	−14.31	−0.705	**
	BMI	−0.240	−13.26	−0.667	**
	WHR	−0.212	−5.95	−0.582	**
	BAI	−0.208	−6.79	−0.578	**
	Systolic blood pressure	−0.110	−2.76	−0.303	*
	Glucose	−0.306	−10.90	−0.837	**
	C-peptide	−0.064	−3.07	−0.172	**
	Insulin	−0.073	−4.62	−0.239	**
	Proinsulin	−0.143	−2.38	−0.388	*
	HOMA R	−0.160	−10.25	−0.468	**
	HOMA F	0.158	7.53	0.391	**
	Alanine aminotransferase (ALT)	−0.218	−8.23	−0.595	**
	AST/ALT ratio	0.257	10.51	0.704	**
	Free thyroxine (fT4)	−0.173	−3.18	−0.473	**
	Free triiodothyronine (fT3)	−0.205	−6.35	−0.563	**
	Uric acid (UA)	−0.130	−4.17	−0.359	**
	Creatinine	0.105	2.14	0.290	*
	Sex hormone-binding globulin (SHBG)	0.206	6.95	0.563	**
	17-Hydroxypregnenolone	0.114	2.45	0.319	*
	DHEA	0.081	1.93	0.211	*
	Androstenediol sulfate	−0.094	−2.41	−0.260	*
	Progesterone	−0.197	−4.72	−0.522	**
	17-Hydroxyprogesterone	0.181	8.21	0.493	**
	Cortisol	0.183	4.18	0.501	**
	5α-Dihydrotestosterone	−0.101	−6.87	−0.271	**
	Estradiol	−0.161	−3.91	−0.442	**
	Allopregnanolone sulfate	−0.095	−2.84	−0.261	*
	Isopregnanolone	−0.085	−2.31	−0.232	*
	Conjugated 5β-pregnane-3α,20α-diol	0.079	2.12	0.212	*
	Epiandrosterone	0.070	2.83	0.194	*
	5α-Androstane-3α,17β-diol	−0.188	−4.27	−0.532	**
	Conjugated 5α-androstane-3α,17β-diol	−0.135	−4.18	−0.372	**
	Conjugated 5α-androstane-3β,17β-diol	−0.125	−3.13	−0.346	**
	A-D+ vs. A+D- group (LLR)	1.000	48.95	0.898	**
**Explained variability**		80.7% (72.9% after cross-validation)			
Sensitivity = 0.963 (0.892, 1.034), Specificity = 0.978 (0.936, 1.02)					

R: Component loadings expressed as correlation coefficients with a predictive component, * *p* < 0.05, ** *p* < 0.01.

**Table 10 ijms-24-08575-t010:** Relationships between 33 men with AD (A+D-) vs. 25 men with T2DM (A-D+). Logarithm of likelihood ratio (LLR) and relevant predictors, as evaluated by the OPLS model (for details, see the Section 4.7).

	Variable	Component Loading	t-Statistic	R	
Relevant predictors (matrix **X**)	Age	0.157	7.44	0.413	**
	Abdominal circumference	−0.163	−2.75	−0.428	*
	Hip circumference	−0.206	−3.65	−0.549	**
	Waist circumference	−0.184	−3.72	−0.481	**
	BMI	−0.156	−3.45	−0.460	**
	WHR	−0.092	−1.79	−0.245	
	BAI	−0.126	−3.52	−0.344	**
	Glucose	−0.298	−9.06	−0.787	**
	Proinsulin	−0.177	−3.53	−0.464	**
	HOMA R	−0.075	−1.79	−0.176	
	HOMA F	0.207	4.60	0.567	**
	Triacylglycerols	−0.138	−4.14	−0.361	**
	Alanine aminotransferase (ALT)	−0.203	−2.19	−0.534	*
	AST/ALT ratio	0.216	3.01	0.568	**
	Uric acid (UA)	−0.163	−3.12	−0.432	**
	Sex hormone-binding globulin (SHBG)	0.204	4.14	0.521	**
	17-Hydroxypregnenolone	−0.140	−2.67	−0.373	*
	17-Hydroxyprogesterone	0.186	2.63	0.484	*
	Cortisol	0.119	2.04	0.303	*
	Androstenediol sulfate	−0.132	−2.57	−0.347	*
	Androstenedione	−0.144	−2.45	−0.383	*
	5α-Dihydrotestosterone	0.093	2.08	0.250	*
	Estradiol	−0.257	−10.65	−0.691	**
	Allopregnanolone	−0.207	−3.02	−0.557	**
	Isopregnanolone	−0.192	−3.80	−0.518	**
	Isopregnanolone sulfate	−0.148	−5.72	−0.392	**
	Pregnanolone	0.143	2.55	0.385	*
	Conjugated 5β-pregnane-3α,20α-diol	0.144	2.97	0.380	*
	Androsterone	−0.159	−3.47	−0.425	**
	Etiocholanolone	0.090	1.50	0.237	
	5α-Androstane-3α,17β-diol	−0.211	−10.07	−0.562	**
	Conjugated 5α-androstane-3α,17β-diol	−0.175	−2.17	−0.464	*
	Conjugated 5α-androstane-3β,17β-diol	−0.206	−4.16	−0.547	**
	Conjugated 5β-androstane-3α,17β-diol	−0.133	−1.46	−0.350	
	A-D+ vs. A+D- group (LLR)	1.000	18.70	0.921	**
**Explained variability**		84.8% (77.4% after cross-validation)			
Sensitivity = 1 (1, 1), Specificity = 1 (1, 1)					

R: Component loadings expressed as correlation coefficients with a predictive component, * *p* < 0.05, ** *p* < 0.01.

**Table 11 ijms-24-08575-t011:** Relationships between 7 women with AD and T2DM (A+D+) vs. 41 women with AD (A+D-). Logarithm of likelihood ratio (LLR) and relevant predictors, as evaluated by the OPLS model (for details, see the Section 4.7).

	Variable	Component Loading	t-Statistics	R	
Relevant predictors (matrix **X**)	WHR	0.057	0.96	0.149	
	Aspartate aminotransferase (AST)	−0.221	−3.08	−0.603	**
	Creatinine	0.027	0.38	0.074	
	Pregnenolone	−0.166	−3.11	−0.448	**
	16α-Hydroxypregnenolone	−0.286	−6.61	−0.781	**
	Dehydroepiandrosterone (DHEA)	−0.322	−8.87	−0.881	**
	7α-Hydroxy-DHEA	−0.325	−11.44	−0.888	**
	Androstenediol sulfate	−0.233	−4.15	−0.637	**
	5-Androstene-3β,7β,17β-triol	−0.298	−7.12	−0.815	**
	Isopregnanolone	−0.299	−11.98	−0.817	**
	Conjugated 5β-pregnane-3α,20α-diol	−0.211	−3.89	−0.575	**
	Androsterone	−0.287	−8.35	−0.786	**
	Epiandrosterone	−0.299	−9.38	−0.817	**
	Etiocholanolone	−0.295	−5.93	−0.808	**
	Etiocholanolone sulfate	−0.279	−5.99	−0.765	**
	Conjugated 5α-androstane-3β,17β-diol	−0.257	−4.36	−0.702	**
	Conjugated 5β-androstane-3α,17β-diol	−0.258	−5.79	−0.709	**
	A+D+ vs. A+D- group (LLR)	1.000	5.95	0.525	**
**Explained variability**		27.6% (17.6% after cross-validation)			
Sensitivity = 0.167 (−0.132, 0.465), Specificity = 1 (1, 1)					

R: Component loadings expressed as correlation coefficients with a predictive component, ** *p* < 0.01.

**Table 12 ijms-24-08575-t012:** Relationships between 7 men with AD and T2DM (A+D+) vs. 33 men with AD (A+D-). Logarithm of likelihood ratio (LLR) and relevant predictors, as evaluated by the OPLS/MR model (for details, see the Section 4.7).

	Variable	Component Loading	t-Statistic	R	
Relevant predictors (matrix **X**)	Abdominal circumference	0.283	3.56	0.543	**
	Glucose	0.320	1.43	0.603	
	C-peptide	−0.045	−0.47	−0.085	
	Creatinine	−0.275	−3.76	−0.472	**
	Pregnenolone	−0.360	−4.15	−0.683	**
	16α-Hydroxypregnenolone	−0.388	−2.86	−0.740	*
	20α-Dihydropregnenolone	−0.357	−2.32	−0.682	*
	Androstenediol	−0.275	−1.87	−0.527	
	Progesterone	−0.316	−6.12	−0.600	**
	17-Hydroxyprogesterone	−0.279	−5.33	−0.533	**
	Isopregnanolone	−0.355	−4.33	−0.675	**
	Etiocholanolone	−0.284	−4.68	−0.542	**
	A+D+ vs. A+D- group (LLR)	1.000	7.52	0.682	**
**Explained variability**		46.5% (37.5% after cross-validation)			
Sensitivity = 0.429 (0.062, 0.795), Specificity = 1 (1, 1)					

R: Component loadings expressed as correlation coefficients with a predictive component, * *p* < 0.05, ** *p* < 0.01.

**Table 13 ijms-24-08575-t013:** Relationships between 7 women with AD and T2DM (A+D+) vs. 47 women with T2DM (A-D+). Logarithm of likelihood ratio (LLR) and relevant predictors, as evaluated by the OPLS model (for details, see the Section 4.7).

	Variable	Component Loading	t-Statistic	R	
Relevant predictors (matrix **X**)	Age	0.205	5.79	0.484	**
	Abdominal circumference	−0.088	−2.37	−0.104	*
	Hip circumference	−0.179	−4.12	−0.338	**
	Waist circumference	−0.131	−2.84	−0.212	*
	BMI	−0.151	−2.43	−0.326	*
	BAI	−0.198	−6.01	−0.412	**
	Glucose	−0.163	−2.05	−0.432	*
	Proinsulin	−0.119	−2.36	−0.240	*
	Free fatty acids	0.205	2.94	0.501	*
	Alanine aminotransferase (ALT)	−0.256	−5.41	−0.600	**
	Aspartate aminotransferase (AST)	−0.138	−2.63	−0.310	*
	AST/ALT ratio	0.263	4.22	0.621	**
	Free thyroxine (fT4)	−0.221	−2.00	−0.555	*
	Uric acid (UA)	−0.187	−2.71	−0.443	*
	Pregnenolone	−0.127	−6.40	−0.378	**
	16α-Hydroxypregnenolone	−0.136	−4.76	−0.397	**
	7α-Hydroxy-DHEA	−0.157	−7.19	−0.457	**
	Androstenediol sulfate	−0.163	−3.91	−0.393	**
	5-Androstene-3β,7α,17β-triol	−0.115	−3.56	−0.336	**
	5-Androstene-3β,7β,17β-triol	−0.174	−7.53	−0.503	**
	Progesterone	−0.171	−3.10	−0.501	**
	17-Hydroxyprogesterone (RIA)	0.079	2.75	0.197	*
	Androstenedione	−0.102	−3.65	−0.299	**
	Estrone	−0.156	−4.83	−0.451	**
	Estradiol	−0.166	−2.51	−0.482	*
	Isopregnanolone	−0.215	−6.09	−0.619	**
	Isopregnanolone sulfate	−0.167	−4.17	−0.413	**
	Pregnanolone	−0.149	−1.94	−0.409	*
	Conjugated pregnanolone	−0.142	−2.81	−0.375	*
	Androsterone	−0.080	−2.18	−0.242	*
	Androsterone sulfate	−0.123	−3.20	−0.335	**
	Epiandrosterone	−0.128	−4.08	−0.378	**
	Epiandrosterone sulfate	−0.134	−3.80	−0.369	**
	5α-Androstane-3α,17β-diol	−0.222	−6.68	−0.639	**
	Conjugated 5α-androstane-3α,17β-diol	−0.155	−3.04	−0.391	**
	Conjugated 5α-androstane-3β,17β-diol	−0.178	−3.95	−0.443	**
	Conjugated 5β-androstane-3α,17β-diol	−0.168	−2.15	−0.421	*
	A+D+ vs. A-D+group (LLR)	1.000	17.82	0.864	**
**Explained variability**		74.6% (60.5% after cross-validation)			
Sensitivity = 1 (1, 1), Specificity = 1 (1, 1)					

R: Component loadings expressed as correlation coefficients with a predictive component, * *p* < 0.05, ** *p* < 0.01.

**Table 14 ijms-24-08575-t014:** Relationships between 7 men with AD and T2DM (A+D+) vs. 25 men with T2DM (A-D+). Logarithm of likelihood ratio (LLR) and relevant predictors, as evaluated by the OPLS model (for details, see the Section 4.7).

	Variable	Component Loading	t-Statistic	R	
Relevant predictors (matrix **X**)	Age	0.187	3.46	0.457	**
	Hip circumference	−0.141	−2.09	−0.345	*
	BMI	−0.193	−2.51	−0.480	*
	BAI	−0.167	−2.31	−0.410	*
	C-peptide	−0.275	−2.97	−0.675	*
	Total cholesterol	−0.193	−2.46	−0.475	*
	LDL cholesterol	−0.168	−2.80	−0.414	*
	Alanine aminotransferase (ALT)	−0.236	−3.13	−0.562	**
	AST/ALT ratio	0.218	2.38	0.520	*
	Thyrotropin (TSH)	−0.177	−3.41	−0.434	**
	Uric acid (UA)	−0.207	−3.33	−0.492	**
	Creatinine	−0.206	−3.78	−0.470	**
	C-reactive protein (CRP)	−0.099	−1.72	−0.239	
	Pregnenolone	−0.206	−3.94	−0.538	**
	Androstenediol	−0.218	−3.88	−0.569	**
	Progesterone	−0.252	−5.06	−0.658	**
	Allopregnanolone	−0.292	−8.63	−0.763	**
	Isopregnanolone	−0.170	−3.21	−0.444	**
	Isopregnanolone sulfate	−0.252	−3.64	−0.616	**
	Pregnanolone	0.116	2.36	0.302	*
	Conjugated pregnanolone	−0.224	−3.38	−0.547	**
	Androsterone	−0.240	−5.57	−0.626	**
	Epiandrosterone	−0.220	−5.45	−0.574	**
	5α-Androstane-3α,17β-diol	−0.254	−2.45	−0.665	*
	A+D+ vs. A-D+ group (LLR)	1.000	10.93	0.781	**
**Explained variability**		60.9% (48.3% after cross-validation)			
Sensitivity = 0.833 (0.535, 1.132), Specificity = 0.96 (0.883, 1.037)					

R: Component loadings expressed as correlation coefficients with a predictive component, * *p* < 0.05, ** *p* < 0.01.

## Data Availability

Original data presented in the study are available upon reasonable request to the corresponding autors. The data are not publicity available due to privacy policy.

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
