# Peer review of "The Role of Steroidomics in the Diagnosis of Alzheimer’s Disease and Type 2 Diabetes Mellitus"

_ijms, 2023, doi:10.3390/ijms24108575_

Round 1

Reviewer 1 Report

In this research, the authors attempt to compare and contrast anthropometric and steroid measures in AD and T2DM to separate the diseases metabolically. Given the considerable overlap of disrupted metabolic pathways between the two diseases, this topic is an interesting and valid research question. 

However, by trying to adjust for diabetes, AD, and sex in the analysis, the authors have limited the group sizes and radically increased the complexity of the paper. It appears that the initial study design was meant to analyze AD and T2DM differences, and this split by sex was added as a secondary analysis. I assume this because of the radically different group sizes between men and women and the very small group sizes for the T2DM+, AD+ groups. Also, I would have to assume that if the analysis were done without splitting the data between male and female, few significant results would be found since this analysis is not mentioned in the manuscript.

The sample sizes for the T2DM+, AD+ groups were considerably smaller than the others. This sample size could and likely does cause a biased result. The potential bias should be acknowledged in the paper with the effects on the statistical analysis described and the caveats discussed in the conclusions. I wondered how many of the differences were due to actual disease differences and how many were artifacts from the small group sizes. 

The authors spent considerable effort describing each molecule difference between the eight groups for each table, yet there are very few overarching comparisons. As an example, Tables 11-14 describe the impact of diabetes. Yet, if you look at the results, the results for men and women without AD (T11-12) are relatively similar. The results with AD (T13-14) show apparent differences from those without AD. It might be better to combine Tables 11 and 12 and to combine Tables 13 and 14. The combined tables would make it easier to compare men vs. women. The tables would also make comparing and contrasting the differences seen with and without AD easier. Also, it might be interesting to see the results without the analysis split by sex. A summation of pathway differences in T2DM with and without AD would be a good addition. The same comparison situation holds for Tables 7-10. Combining the tables could make it easier to see the differences and similarities and read the document. These changes would support the introduction with its initial goals. “In this study, we were primarily interested in the extent to which AD and T2DM are similar and different, and whether these pathologies are related only to aging or whether they are synergistic diseases associated with interacting pathophysiological cycles, and if AD and T2DM might occur in comorbidity.”

Fasting glucose and insulin measurements are known to be highly variable. This variability is why most individuals with T2DM are evaluated using an estimated average glucose or A1c measure. It is unclear from the results if the T2DM+, AD- group (both male and female) had worse glucose control than the T2DM+, AD+ group due to disease, just worse glucose control for the blood draw day, or if there was a sample bias. The T2DM+, AD- also had higher BMI than the T2DM+, AD+ group. Again, whether this is a disease effect or sample bias is unclear. Sections 2.1.1. and 2.1.2. contains the results and correlation of glucose and obesity-related measures with the T2DM+, AD- group showing almost exactly what would be predicted offhand based on the differences in BMI and fasting glucose independent of AD. I don’t think you can conclude (section 3.1.1) that the T2DM+, AD+ is underweight or has better glucose control given the small sample size and lack of A1c measure. Providing an A1c/estimated average glucose could show whether there were differences in diabetes disease control over the short term, perhaps allowing a stronger statement regarding glucose metabolism and AD. 

Are the paragraphs starting with line 1738 repeating the paragraph starting with line 1731? And did the subjects with T2DM undergo or not undergo the same test protocols (lines 1736 and 1738)?

Line 1750, do you mean treated for diabetes or depression and treated with anti-hypertensives?

The text states: “Anthropometric and biochemical parameters are shown in Tables 1, S1, and S2, sex hormone-binding globulin (SHBG) and steroids are shown in Tables 2, S3, and S4, and ratios of products and precursors that may reflect steroidogenic enzyme activities and the balance between these activities are shown in Tables 3, S5, and S6. Tables S7-S10 show the results of the OPLS model assessing the effect of AD, Tables S11-S14 show the effect of T2DM, and Tables S15 and S16 show the differences between patients with AD and without T2DM on the one hand and patients with T2DM and without AD on the other.” This list mentions a total of 19 tables.

I found 16 tables labeled Table 1-16. Anthropometric and biochemical parameters were in Tables 1 and 2, Steroids were in Tables 3 and 4, Ratios in Tables 5 and 6, and Model results in Tables 7-16. So there appear to be three missing tables and six tables where the numbers in the text don’t match the table numbers. 

Please check the grammar and punctuation. 

This manuscript is long and complex. The introduction for this manuscript is a reasonable length and contains 14 references. The methods are an appropriate length and include six references. The discussion section is 22.5 pages long and contains 178 references with two pages of conclusions following the Methods. This discussion is longer than most review papers. I would consider it unreasonably long. The discussion section goes molecule by molecule, with a review section and a sentence about the results. For example, paragraphs (3.2.1) on SHBG contain 24 lines and 16 references describing prior literature, with three lines that restate the results and conclude the results were consistent with the prior references. Consider deleting all but two pages of the discussion and make the conclusions consistent with the discussion section. 

The research in this manuscript is interesting and could be a nice publication. Unfortunately, the manuscript needs significant work and editing. 

Reviewer 2 Report

The authors presented a study outlining the relationship between steroids and Alzheimer`s disease and T2D. While the manuscript has several strengths, there are few issues that need to be addressed:

11. I suggest a larger paragraph with the relation between T2D and steroids in the “Introduction” section.

22.   For this study, I believe the treatment used for T2D patients and AD patients is very important in the context of circulating steroids and other characteristics, think about it, maybe a potential division in group of treatments should be tried. Does all these patients have other medical history or other associated diseases? These needs to be mentioned, also.

33.     The age is another parameter that may influence the results, as you already mentioned the difference of age in the groups of investigated patients, to be introduced in ”Study limitations section.

Round 2

Reviewer 1 Report

The authors have put considerable work into shortening the manuscript. The edits have improved the document's readability considerably and focused the discussion on the results of this research.

 A significant portion of this manuscript examines the differences between men and women with and without AD and T2DM. The sex differences are the focus of all tables, methods, discussion, and conclusions. Neither the abstract or introduction mention this crucial part of the research. I recommend altering the last sentence in the introduction paragraph (starting at line 68) to say that the analysis also compares males vs. females. Also, consider updating the abstract to mention the work includes AD vs. T2DM and males vs. females.

Author Response

The authors are very grateful to reviewer 1 for his comments. All requests and comments of reviewer #1 have been addressed in the new version.